# The effect of enantioselective chiral covalent organic frameworks and cysteine sacrificial donors on photocatalytic hydrogen evolution

Weijun Weng[1] & Jia Guo [1]✉

Covalent organic frameworks (COFs) have constituted an emerging class of organic photocatalysts showing enormous potential for visible photocatalytic $H_2$ evolution from water. However, suffering from sluggish reaction kinetics, COFs often cooperate with precious metal co-catalysts for essential proton-reducing capability. Here, we synthesize a chiral β-ketoenamine-linked COF coordinated with 10.51 wt% of atomically dispersed Cu(II) as an electron transfer mediator. The enantioselective combination of the chiral COF-Cu(II) skeleton with L-/D-cysteine sacrificial donors remarkably strengthens the hole extraction kinetics, and in turn, the photoinduced electrons accumulate and rapidly transfer via the coordinated Cu ions. Also, the parallelly stacking sequence of chiral COFs provides the energetically favorable arrangement for the H-adsorbed sites. Thus, without precious metal, the visible photocatalytic $H_2$ evolution rate reaches as high as 14.72 mmol $h^{-1}$ $g^{-1}$ for the enantiomeric mixtures. This study opens up a strategy for optimizing the reaction kinetics and promises the exciting potential of chiral COFs for photocatalysis.

Photocatalytic hydrogen evolution (PHE) holds great promise to generate clean fuels from renewable sunlight resources. The process of solar-to-hydrogen transduction relies on the photocatalysts, which serve as a photosensitizer to harvest solar energy and as an electron relay to manage proton reduction[1]. To elevate $H_2$ production efficiency, sacrificial electron donors (SEDs) are utilized with photocatalysts in water splitting for hole extraction. This scavenging process causes a strong asymmetry in charge carrier kinetics, by which the holes are trapped for electron accumulation[2,3]. Thus, the photooxidation kinetics with SEDs is essential in photogenerated charge separation for $H_2$ generation. As is well known from enzyme catalysis in nature, the enantioselective reactions are beneficial to reinforcing the kinetics by selectively binding chiral substrates with enzyme[4]. More intriguingly, chiral-induced spin selectivity and polarization contribute to elevating the intermediate reactivity and selecting the desired reaction pathways, e.g., oxygen evolution reaction in water

electrolysis[5] and photocatalytic water splitting under circularly polarized light[6]. It motivates us to envision the enantioselective oxidation of SEDs with chiral organic photocatalysts to increase PHE kinetics for enhanced photocatalytic activity.

Amongst the organic photocatalysts, two-dimensional covalent organic frameworks (2D COFs) are one of the most promising materials, which feature a coaxially stacked planar framework with designable topological diagrams[7–9], adaptative chemical functionality[10–12], and tunable porous properties[13–15]. Besides the photosynthetic applications, tremendous efforts on 2D COFs have been devoted to various fields such as separation[16,17], sensing[18,19], catalysis[20,21], ion conduction[22,23], energy storage[24,25], and electronic devices[26,27]. Meanwhile, 2D COFs have established an unequivocal structure-to-activity correlation based on periodic atomic frameworks and ordered stacking sequences. Especially for photosynthesis, the crystalline 2D COFs exhibit remarkable superiority compared to the other known

[1]State Key Laboratory of Molecular Engineering of Polymers, Department of Macromolecular Science, Fudan University, Shanghai, China.
✉e-mail: guojia@fudan.edu.cn

amorphous organic photocatalysts[28–30]. The photocatalytic functionality has been promoted by incorporating photosensitizing groups in building blocks[31,32], constructing donor-acceptor on skeletons[33–35], and extending π-electron conjugation along backbones[36–38]. Consequently, COF-related photocatalysts are bestowed with broad-band absorptivity and high photogenerated charge mobility. Nevertheless, utilization of COFs alone is inadequate for PHE due to the high overpotential of proton reduction and coupling. Thus, precious metals such as Pt and Pd need to exist as a co-catalyst, lowering the overpotentials of $H_2$ production and enhancing photo-induced electron accumulation[39,40]. Unambiguously, without precious metal co-catalysts, the intrinsic photocatalytic performances of 2D COFs are far from satisfactory in the $H_2$ evolution rate while remain unexplored so far.

In recent years, chirality has been successfully imparted to 2D COFs by equipping chiral centers into the backbones[41,42] or side groups of skeletons[43,44]. They can implement the chiral recognition[45] and catalyze the enantiomeric substrates with high efficiency and selectivity[46,47]. These studies imply that the chiral 2D COFs are likely to expedite the redox kinetics via the enantioselective combination with SEDs. With the concept of artificial enzymes in mind, we synthesize a chiral β-ketoenamine-linked COF for exploring an enantiomeric system for visible PHE. By using L-/D-cysteine as a SED and Cu ion as an electron transfer mediator, the chiral 2D COF can directly generate a significant flux of $H_2$ gas upon exposure to visible irradiation (>420 nm). Briefly, Cu(II) ions are immobilized at the chiral center with N-salicylideneaniline subunits to afford atomically dispersed sites on the COF skeletons. As L-/D-cysteine is selectively bonded with the chiral COF, the SED oxidation is dramatically enhanced via a circular reaction, by which cysteine reduces the coordinated Cu(II) into Cu(I) and subsequently the photo-induced holes oxidize Cu(I) to Cu(II). Meanwhile, the chiral crystalline frameworks exhibit a smaller overpotential in $H_2$ production than the achiral analogs, implying the lowered energy barriers and enhanced catalytic activity. Therefore, thanks to the intensified reaction kinetics on the SED oxidation and proton reduction, the enantioselective combination of COF and cysteine exhibits a record high $H_2$ evolution rate (HER) of 14.72 mmol $h^{-1}$ $g^{-1}$ and sacrificial oxidation turnover frequency ($TOF_{ox}$) of 9.0 $h^{-1}$. The HER is comparable to those of many achiral COFs using Pt as a co-catalyst. Our study would open up a promising avenue to exploit the potential of chiral organic materials for solar-to-hydrogen photocatalysis.

## Results

### Immobilization of single-atom copper on COFs

As a proof of concept, we embarked on studying the photocatalytic behaviors of the achiral COFs without precious metals because PHE is still challenging for the organic photocatalysts. The COF materials were modified by complexing with Cu(II) ions for improving the electronic properties of organic components. A series of achiral β-ketoenamine-linked COFs were solvothermally synthesized by the aldimine condensation of 1,3,5-triformylphloroglucinol (Tp) and diamine-substituted linkers with the specific lengths varied by phenyl, biphenyl and terphenyl groups, respectively. The subsequent irreversible enol-to-keto transformation resulted in the β-ketoenamine linkage, consisting of keto (C=O) and enamine (NH) with the intramolecular H-bonding[48]. Through the rigorous characterizations, the various COFs possessed the characteristic hexagonal lattice structures, high surface areas, and uniform pore-size distributions, demonstrating their periodic planar frameworks and ordered stacking structures (Supplementary Figs. 1, 2). Also, the β-ketoenamine formation was convincingly validated by a series of molecular spectroscopies (Supplementary Figs. 3, 4).

Then, the complexation of Cu ions with the COF backbones proceeded in the mild conditions by adding $Cu(OAc)_2$ into the aqueous dispersion of COF solid. As shown in Fig. 1a, the Cu(II) coordination

occurs on a bidentate ligand with phenolic oxygen and imine nitrogen, derived from the reversed keto-to-enol transformation. The TpPa-COF containing phenyl as a linker was applied as a representative to investigate the coordination structure of the resulting product, marked as TpPa-Cu(II)-COF. As a control, a model compound TpPa-Cu(II) was synthesized by the reaction of Tp and aniline and complexed with Cu(II) ions under similar conditions. A variety of spectroscopic techniques were performed to identify the TpPa-Cu(II) complexes. The vibration band of C=N was observed at 1601 $cm^{-1}$, which was coordinated with Cu(II) to induce a slight shift to the low wavenumbers[49]. Also, the C=O peak at 1620 $cm^{-1}$ originating from the TpPa model nearly disappeared in the spectrum of the TpPa-Cu(II) model (Supplementary Fig. 5). For the TpPa-Cu(II)-COF, the analogous change was observed that the vibration intensity of the C=O band was weakened gradually with an increase in the coordinated Cu(II) ions. X-ray photoelectron spectroscopy (XPS) provided evidence for metal valence and atomic binding energy. As displayed in Fig. 1b, the Cu 2p XPS spectra of the TpPa-Cu(II)-COF and TpPa-Cu(II) model exhibit a dominant peak of Cu $2p_{3/2}$ at 933.9 eV with its adjacent satellites, which is slightly shifted by 0.7 eV compared with that of copper acetate (934.6 eV). The N 1s core-level spectrum of TpPa-Cu(II)-COF could be deconvoluted into the two peaks at 399.4 and 398.5 eV (Fig. 1c). In contrast to the XPS spectra of the parent TpPa-COF and TpPa-Cu(II) model, the peak at 399.4 eV could be attributed to the C–N bond originating from the β-ketoenamine linkage[50]. The other emerging peak at 398.5 eV was ascribed to the newly formed C=N bond, which suggested the occurrence of the keto-to-enol transformation. We assume that the reversed tautomerization is caused by the metal complexation, which leads to the electron rearrangement on the ketoenamine bonds for the strong affinity towards Cu(II). Hence, the enol-imine instead of ketoenamine tautomer robustly immobilizes Cu(II) ions onto the skeletons of TpPa-COF (Fig. 1d). The quantity of the coordinated Cu(II) ions confirmed by ICP-AES could be controlled from 2.90 to 12.77 wt% (Supplementary Table 1).

Next, the effect of metal coordination on COF structures was studied by powder X-ray diffraction (PXRD) and $N_2$ sorption measurements. The dominant X-ray diffraction peaks of TpPa-Cu(II)-COF(10.76 wt%) could be attributed to the TpPa-COF component (Fig. 1e). Also, the Brunauer–Emmet–Teller surface area of the complex was as high as 759 $m^2$ $g^{-1}$, similar to that of the pristine TpPa-COF (Fig. 1f). The type-I sorption isotherm manifested the micropore character of TpPa-Cu(II)-COF(10.76 wt%). The pore-size distribution derived from the NLDFT method was populated at 1.5 nm (Fig. 1f, inset). All the results validate that the periodic structure with the accessible pore channels well remains after the coordination of Cu(II) ions with TpPa-COF.

The coordination environments of TpPa-Cu(II)-COF were thoroughly dissected with the Cu K-edge measured by the X-ray absorption spectroscopy. To our knowledge, when the X-ray absorption near edge structure spectra (XANES) involve a high binding energy of absorption edge, the coordinated metal center is kept in a high-valence state. Thus, compared to the XANES spectrum of Cu foil, it was confirmed that the coordinated Cu metal remained positively charged in the TpPa-COF (Supplementary Fig. 33a). As shown in Fig. 1g, the extended X-ray absorption fine structure (EXAFS) r-space with the Fourier transformation of k = 3–13 $Å^{-1}$ offers a dominant shell peak for TpPa-Cu(II)-COF at 1.5 Å attributable to the Cu-O/N bonding. In contrast, the second shell peak around 1.9–2.7 Å is relatively weak, which could be assigned to the Cu-C path. In contrast to the EXAFS r-space of Cu foil, TpPa-Cu(II)-COF has few Cu-Cu bonding (2.2 Å), indicating the possibility of single-atom Cu(II) distribution in the TpPa-COF. Again, the Morlet wavelet analysis of k-space was carried out to distinguish the atomic environments (Fig. 1h). The center of the backscattering wave function for TpPa-Cu(II)-COF was positioned in the relatively low k-space. This magnifies that the neighbors to Cu metal are light atoms

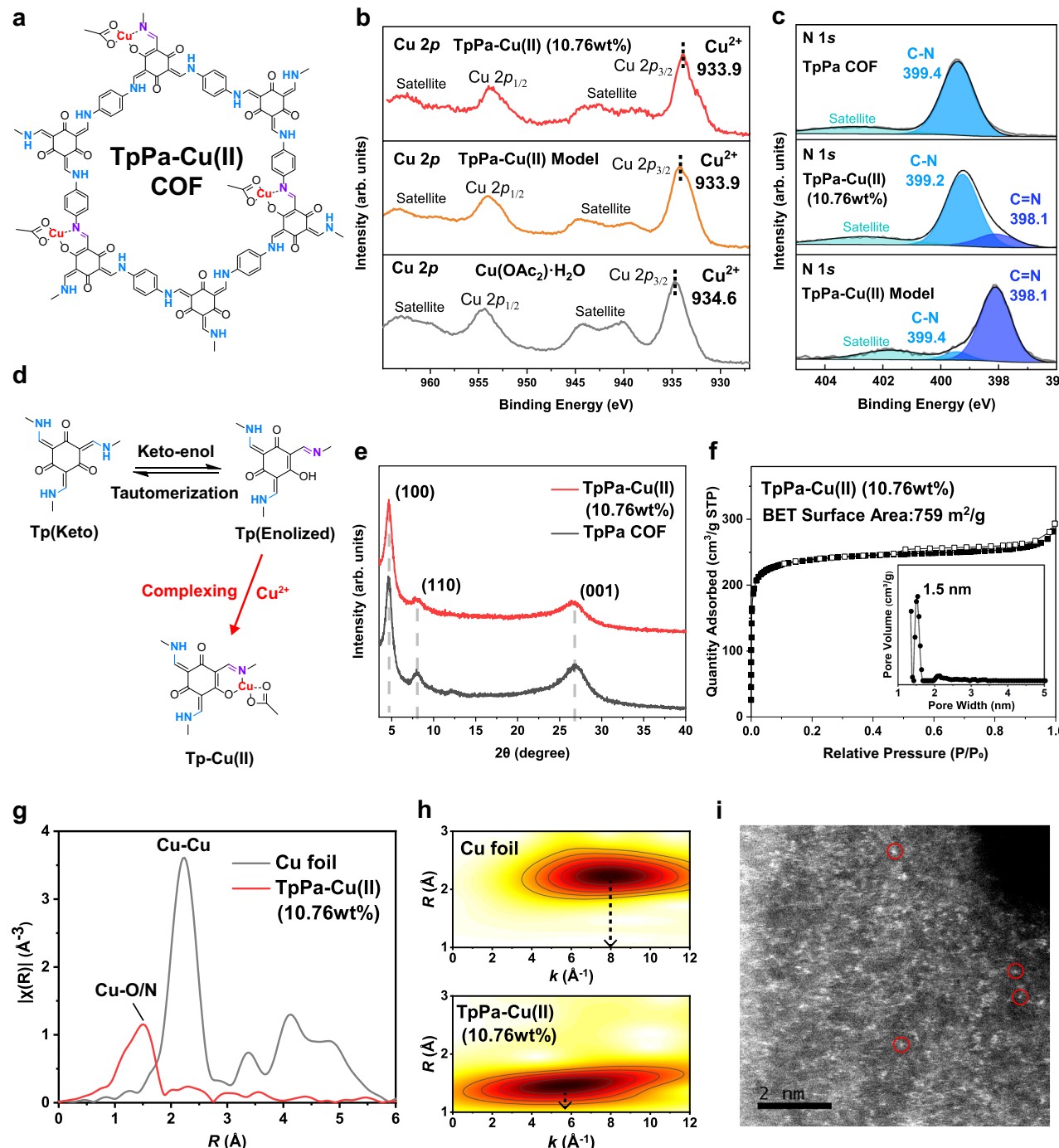

**Fig. 1 | Characterizations of the Cu(II)-coordinated TpPa-COF. a** Structure of the TpPa-Cu(II)-COF complex. **b** Cu 2*p* XPS spectra of TpPa-Cu(II)-COF(10.76 wt%), TpPa-Cu(II) model compound and Cu(OAc)$_2$•H$_2$O. **c** N 1*s* XPS spectra of TpPa-COF, TpPa-Cu(II)-COF(10.76 wt%) and TpPa-Cu(II) model compound. **d** Structural tautomerization of keto to enol for coordination of Cu$^{2+}$ with *N*-salicylideneaniline moiety. **e** PXRD patterns of TpPa-COF before and after coordination with Cu(OAc)$_2$. **f** Nitrogen adsorption (solid circle) and desorption (open circle) isotherm profiles and pore-size distribution (inset) of the TpPa-Cu(II)-COF(10.76 wt%). **g** The r-space distributions and **h** the wavelet analysis calculated from the $k^2$-weighted Cu K-edge EXAFS spectra of TpPa-Cu(II)-COF(10.76 wt%) and Cu foil (without phase correction). **i** HADDF-STEM image of the atomically dispersed Cu in the TpPa-Cu(II)-COF(10.76 wt%). Source data are provided as a Source Data file.

rather than heavy metal atoms. The HAADF-STEM image offered more intuitive evidence that the Cu ions were atomically dispersed without the formation of clusters or nanoparticles (Fig. 1i and Supplementary Fig. 6). In a large view field, there were also no metal nanoparticles in the HR TEM image (Supplementary Fig. 7). The corresponding EDX mapping in the same region disclosed that the high content of Cu(II) metal was uniformly distributed on the COF matrix. Note that 10.76 wt % of atomically dispersed Cu ions in the COF has rarely been reported so far.

## Mechanistic insight into the SED redox reaction

Prior to the photocatalytic test, the energy-band structure of TpPa-Cu(II)-COF was investigated through Mott−Schottky plots and UV−vis−DRS spectra (Supplementary Figs. 8, 9). The optical bandgaps were obtained from the Tauc plots (Supplementary Fig. 10), and the flat band levels determined by the Mott-Schottky curve were approximately regarded as the bottom level of conductive band (CB). Compared with the TpPa-COF, the CB and valence band (VB) of TpPa-Cu(II)-COF(10.76 wt%) slightly shifted from −4.01 to −4.09 eV and from

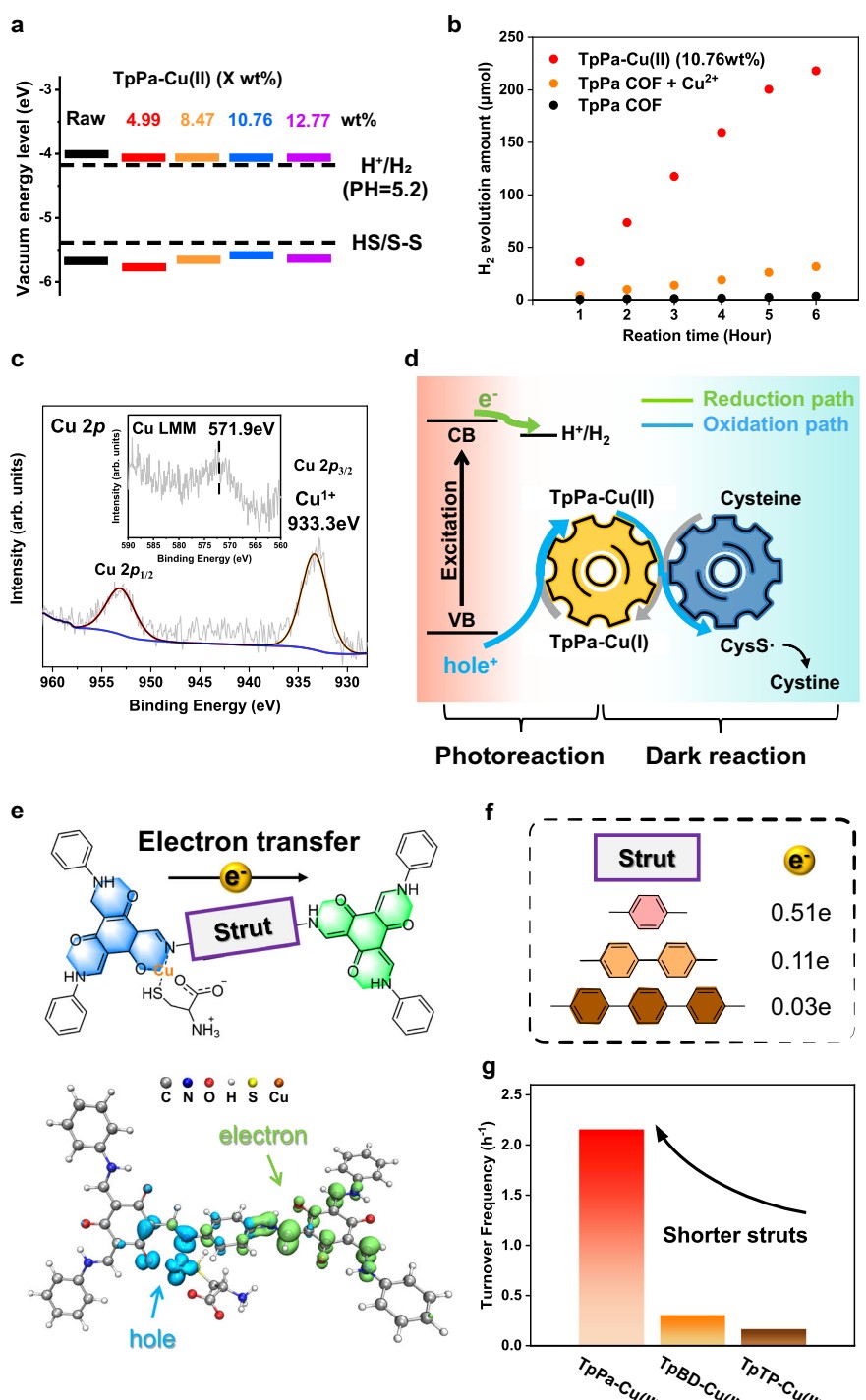

**Fig. 2 | Energy-band structures, photocatalytic H2 evolution, XPS spectra, proposed mechanism, and excited-state simulation. a** Energy-band structures of TpPa-COF and TpPa-Cu(II)-COFs containing 4.99, 8.47, 10.76, and 12.77 wt% of Cu²⁺ ions, respectively. **b** Time courses for photocatalytic H₂ evolution under visible irradiation for TpPa-COF, TpPa-Cu(II)-COF(10.76 wt%), and a mixture of TpPa-COF and Cu(OAc)₂. **c** Cu 2p XPS and Cu LMM auger (inset) spectra of the recycled TpPa-Cu(II)-COF after 1-h photocatalysis. **d** Illustration of photocatalytic mechanism including proton reduction and Cu-mediated cysteine oxidation. **e** Top: theoretical prediction of photogenerated electron pathway. Bottom: real-space distribution of photo-induced electrons (green regions) and holes (blue regions) on the calculated TpPa-Cu(I) model. **f** Calculation of the transferred electron numbers in the cutout models with different linkers. **g** Turnover frequency of photocatalytic H₂ evolution *vs.* the linker lengths of the different Tp-based COFs. Source data are provided as a Source Data file.

−5.68 to −5.78 eV, respectively, and were similar to the others chelating Cu ions of 4.99 wt%, 8.47 wt% and 12.77 wt%, respectively (Fig. 2a). Thus, from the thermodynamic point of view, all the TpPa-Cu(II)-COFs possess the sufficient redox ability for proton reduction (−4.19 eV, pH = 5.2) and cysteine oxidation (−5.41 eV, pH = 5.2) under visible irradiation.

Photocatalytic H₂ evolution in water was carried out on the TpPa-Cu(II)-COF photocatalyst upon exposure to visible irradiation (λ > 420 nm), using L-cysteine as the SED. As no metal co-catalyst was added for H₂ evolution, the TpPa-COF component was the key to the photo-induced electron accumulation and proton reduction. As displayed in Fig. 2b, the achieved HER after 6-h illumination is as high as

3.64 mmol h⁻¹ g⁻¹ and the TOF$_{ox}$ is up to 2.2 h⁻¹ for the TpPa-Cu(II)-COF(10.76 wt%) under the optimized conditions (Supplementary Table 2). As a control, the pristine TpPa-COF gave the HER of 0.04 mmol h⁻¹ g⁻¹ in the presence of L-cysteine. When the free Cu²⁺ ions equivalent to the loaded amount (10.76 wt%) were added in the TpPa-COF dispersion, the HER was increased to 0.53 mmol h⁻¹ g⁻¹, while it was only 20% of the HER for the TpPa-Cu(II)-COF(10.76 wt%) under identical conditions. When the coordinated Cu content was varied from 12.77 wt%, 8.47 wt% to 4.99 wt%, the HERs for the corresponding COF complexes were decreased to 2.94, 2.30, and 0.40 mmol h⁻¹ g⁻¹, respectively (Supplementary Fig. 11). Hence, 10.76 wt% of atomically dispersed Cu(II) is optimum both in quantity and distribution for photocatalytic performance. In addition, when TpPa-Cu(II)-COF(10.76 wt%) was exposed to 12-h visible irradiation, a marginal change was observed in structure, composition, and optical property (Supplementary Fig. 12), as well as a majority of Cu (71%) remained in the recycled COF.

The photocatalytic mechanism of TpPa-Cu(II)-COF in the presence of L-cysteine was thoroughly investigated. Accompanied by H$_2$ evolution, L-cysteine was oxidized into the insoluble cystine in water as determined by ¹H NMR and PXRD (Supplementary Fig. 13). It is unambiguous that L-cysteine plays the role of SED to extract the photogenerated hole from the excited COF backbones. The complexed Cu(II) ions may serve as an electron transfer mediator to expedite the SED oxidation reaction. To corroborate our assumption, the TpPa-Cu(II)-COF recycled after 1-h photocatalysis was characterized by XPS. As displayed in Fig. 2c, the Cu 2$p$ spectrum gives the single 2$p_{3/2}$ and 2$p_{1/2}$ peaks at 933.3 and 953.1 eV, respectively, and the central peak of the Cu LMM Auger spectrum appears at 571.9 eV (Fig. 2c, inset). By comparing the binding energy of Cu 2$p_{3/2}$ peak and the kinetic energy of Cu LMM peak, the modified Auger parameter was calculated to identify the valence state of copper. The obtained value (1848.7 eV) was similar to that reported for the Cu(I)-L-cysteine complex (1848.6 eV)[51], manifesting that the coordinated Cu(II) within TpPa-COF is reduced into Cu(I) by L-cysteine. To further rationalize the origin of Cu(I) species, TpPa-Cu(II)-COF was dispersed in the aqueous solution of L-cysteine in the dark for 1 h. As evidenced by the XPS and X-ray excited Auger spectra (Supplementary Fig. 14), a similar Auger parameter of 1848.4 eV was achieved, validating the formation of Cu(I) in the dark condition. Meanwhile, the XPS spectra of the recycled TpPa-Cu(II)-COF involved the S 2$p$ core-level signal, which could be deconvoluted into the two single peaks at 164.0 eV and 165.3 eV ascribed to the S-Cu bonding and its closely spaced spin-orbit component, respectively (Supplementary Fig. 14c). Electron paramagnetic resonance (EPR) spectroscopy was performed to track the radical generation (Supplementary Fig. 15a). Without light irradiation for the mixture of Cu(II) ions and L-cysteine, a weak quadruple signal was detected in the presence of 5,5-dimethyl-1-pyrroline-N-oxide (DMPO). This is responsible for the DMPO-trapped sulfanyl radicals (CysS•) generated by the oxidation of cysteine[51]. Therefore, the findings underpin that the immobilized Cu(II) ions accept one electron from L-cysteine towards the reductive Cu(I) ions in the dark. During the photocatalysis, the identical radical signals were intensified with the prolonged irradiation as the photo-induced hole on the excited TpPa-Cu(II)-COF was accumulated to enhance the circular conversion between Cu(I) and Cu(II) for the cysteine oxidation (Supplementary Fig. 15b).

Next, cyclic voltammetry (CV) was applied to examine the oxidative potential of L-cysteine in the dark (Supplementary Fig. 16). The observed oxidative potential of L-cysteine (0.1 M) was 0.79 V (vs. Ag/AgCl) in the N$_2$-saturated solution at pH 5.2. With the same conditions, the onset potential of L-cysteine was largely shifted to 0.21 V (vs. Ag/AgCl) in the presence of Cu²⁺ ions (2 × 10⁻⁴ M). In contrast, when the TpPa-Cu(II)-COF solid was coated on the electrode, the oxidative potential could continuously decrease to 0.17 V (vs. Ag/AgCl). It discloses that the TpPa-COF component strengthens the coordinated

Cu(II) electrophilicity to oxidize L-cysteine. On the other hand, the cathodic current density of the TpPa-Cu(II)-COF at −1.0 ~ −0.6 V (vs. Ag/AgCl) was similar to that of the TpPa-COF. Nevertheless, both were far lower than that of the Pt-deposited TpPa-COF. As the theoretical proton reduction potential is −0.51 V (vs. Ag/AgCl), the cathodic current density at the more negative potentials reflects the catalytic ability of materials in the electrochemical reaction. It is therefore likely that the catalytic activity of the complexed Cu(II) ions for H$_2$ production is ignorable, while its role in the L-cysteine oxidation is predominated.

Taken all together, the underlying mechanism of photo-induced hole extraction is proposed, as depicted in Fig. 2d. A dark reaction occurs in the oxidation of L-cysteine with the coordinated Cu(II) on TpPa-COF, yielding the reductive Cu(I) species. Upon visible irradiation, electrons transfer from the Cu(I) donor to the photogenerated acceptor on the excited TpPa-COF, accompanied by the conversion of Cu(I) to Cu(II) state. With the circular redox of Cu ions in the two separate steps, the sulfanyl radicals were continuously yielded and coupled into cystine as the side product. Therefore, the current system offers a cascade process, including the dark reduction of Cu(II) to Cu(I) with L-cysteine and the photochemical process for hole extraction with Cu(I).

### Effect of organic skeletons on electron transfer

As the in-situ formed Cu(I) ions reside in the TpPa-COF during the photocatalysis, it is likely that the photogenerated electron transfer is driven by the coordinated metal ions on the organic skeletons. The study commenced with the TD-DFT calculation to specify the distribution of electrons and holes in solvation models (Supplementary Figs. 17–19 and Supplementary Tables 3–5)[52]. Figure 2e displays the difference in the wavefunctions before and after vertical excitation on the model, in which the green region mainly at the Tp moiety signifies the photo-induced electron accumulation and the blue region at the Tp-Cu(I) complex illustrates the photo-induced electron depletion. Therefore, it is ascertained that the excited-state electrons transfer from the Tp-Cu(I) complex to the Tp moiety via the phenyl linker, resulting in the rearrangement of electron-hole distribution in the excited state. Based on the inter-fragment charge transfer (IFCT) method[53], the value of rearranged electrons between fragments was calculated in the Hirshfeld partition to quantify the transferred charges. One can see that 0.51e of the photo-induced electrons flow via the phenyl linker (Fig. 2f), of which 58% electrons originate from the complexed Cu(I) ions and 42% electrons are contributed by N-salicy-lideneaniline at the same Tp-Cu(I) site (Supplementary Table 6). When the linker length was elongated with biphenyl or terphenyl group, the inter-fragment electron redistribution decreased to 0.11e and 0.03e (Supplementary Fig. 20), respectively, suggesting that the photo-excited charge transfer between donor and acceptor was seriously impaired by the linker length.

With the theoretical prediction in mind, TpBD-Cu(II)-COF and TpTP-Cu(II)-COF were synthesized to examine the performances of photocatalytic H$_2$ evolution. As evidenced by the various characterizations (Supplementary Figs. 2, 4), the newly synthesized COFs possessed the high crystallinity and porosity and coordinated the similar contents of Cu(II), i.e., 10.02 wt% for TpBD-Cu(II)-COF and 8.99 wt% for TpTp-Cu(II)-COF (Supplementary Table 1). Under the identical photocatalytic conditions, TpBD-Cu(II)-COF(10.02 wt%) and TpTp-Cu(II)-COF(8.99 wt%) gave the TOF$_{ox}$ values of 0.31 and 0.17 h⁻¹, respectively (Fig. 2g), which were both relatively lower than that of TpPa-Cu(II)-COF(10.76 wt%) (2.16 h⁻¹). This well agrees with the computational analysis of excited-state electron transfer.

### Enantioselective combination for photocatalytic H$_2$ evolution

As the hole extraction and photo-induced charge transfer process benefited from the anchored Cu ions and the shorter phenyl linker, the enantioselective photocatalysis for H$_2$ evolution was investigated with

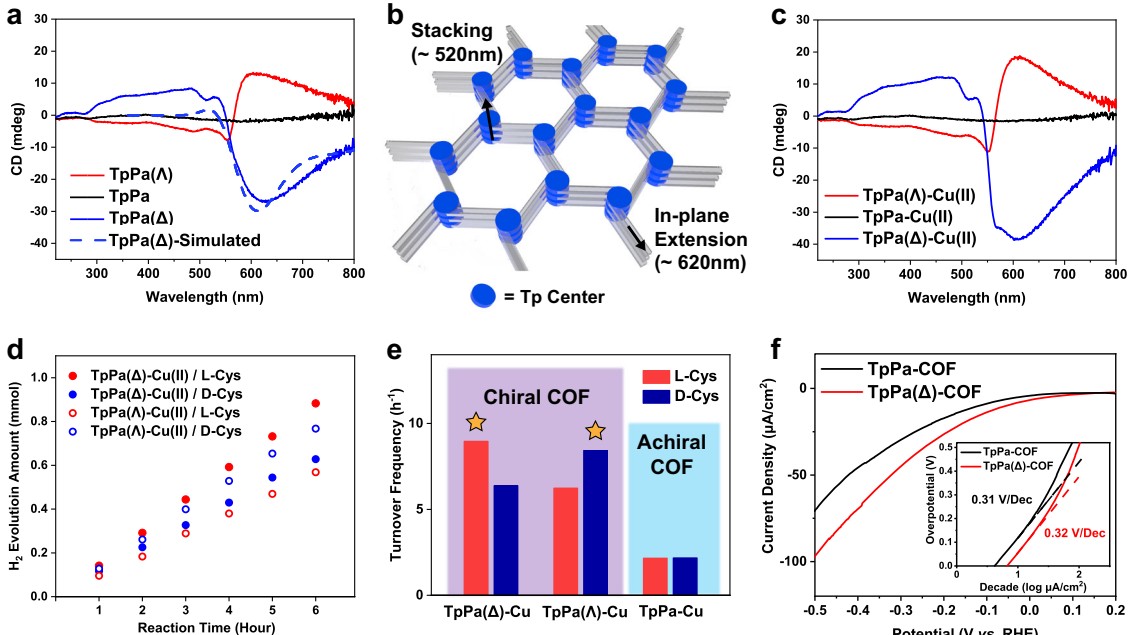

**Fig. 3 | Chiral dichroism spectra, photocatalytic H2 evolution and polarization curves. a** Circular dichroism spectra of the achiral TpPa-COF, TpPa(Δ)-COF, TpPa(Λ)-COF, and double-layered TpPa(Δ) model. **b** Proposed structural response to the circular dichroism signals in the chiral COF. **c** Circular dichroism spectra of the achiral TpPa-Cu(II)-COF, TpPa(Δ)-Cu(II)-COF and TpPa(Λ)-Cu(II)-COF. **d** Time-dependent photocatalytic H₂ evolution and **e** Sacrificial oxidation turnover frequency of the various combinations of TpPa(Δ)-Cu(II)-COF, TpPa(Λ)-Cu(II)-COF and TpPa-Cu(II)-COF with L-/D-cysteine. The stars on the histograms represent the two groups of enantiomeric mixtures. **f** Polarization curves and Tafel curves (inset) of the achiral TpPa-COF and chiral TpPa(Δ)-COF. Source data are provided as a Source Data file.

the optimized TpPa-Cu(II)-COF/cysteine system. Firstly, the chiral TpPa-COF was synthesized by applying a chiral regulator as early reported (Supplementary Figs. 1, 3)[54]. The product exhibited the significant signals derived from the positive and negative Cotton effect in the circular dichroism (CD) spectra (Fig. 3a). To distinguish the TpPa(Δ)-COF and TpPa(Λ)-COF in polarimetry, the corresponding electronic circular dichroism (ECD) spectra were predicted on the level of TD-PBE0-def2SVP//PBE0-def2SVP. The TpPa(Δ)-COF was identified according to the simulated ECD, and its opposite polarized optical behaviors were interpreted via the different COF models (Supplementary Fig. 21). As illustrated in Fig. 3b, the single-layer TpPa(Δ)-COF model implies that the negative Cotton effect at the long wavelength (~620 nm) originates from the in-plane extension of the chiral Tp center. The double-layered model of TpPa(Δ)-COF reveals that the positive Cotton effect at the short wavelength (~520 nm) is derived from the stacking structure of the chiral Tp center. Likewise, such a dual response to circularly polarized light was observed for the TpPa(Λ)-COF but was not detected for the corresponding amorphous polymer (Supplementary Fig. 22).

As validated by the isothermal titration calorimetry, the chiral TpPa-COF could be chelated with Cu ions (Supplementary Fig. 23 and Supplementary Table 7). With the same metal coordination approach, Cu(II) was incorporated into the chiral skeletons, achieving 10.51 wt% for TpPa(Δ)-Cu(II)-COF and 9.72 wt% for TpPa(Λ)-Cu(II)-COF. The chirality, crystallinity, and porosity of the two complexes were all maintained compared to the parent chiral COFs (Fig. 3c and Supplementary Fig. 1). Also, the XPS spectra of Cu 2*p* signals proved that the valence state was identical to that of the achiral TpPa-Cu(II)-COF (Supplementary Fig. 3). The energy-band structure of the chiral TpPa-Cu(II)-COF was similar to the corresponding achiral analog (Supplementary Fig. 24).

Based on the above findings, the photocatalytic test was conducted for the chiral TpPa-Cu(II)-COF in the presence of L-/D-cysteine. Upon exposure to 6-h visible irradiation, the two sets of enantiomeric

mixtures, TpPa(Δ)-Cu(II)-COF/L-cysteine and TpPa(Λ)-Cu(II)-COF/D-cysteine, afforded the HERs of as high as 14.72 and 12.80 mmol h⁻¹ g⁻¹ (Fig. 3d), respectively, which could be ranked among the best of non-precious-metal organic photocatalysts reported so far (Supplementary Table 8). Also, the corresponding TOF$_{ox}$ values increased to 9.0 and 8.5 h⁻¹ for the TpPa(Δ)-Cu(II)-COF/L-cysteine and TpPa(Λ)-Cu(II)-COF/D-cysteine, respectively (Fig. 3e). To validate the reliability of photocatalytic performances, the chiral TpPa(Δ)-Cu(II)-COFs were synthesized in the different batches, exhibiting the similar HERs under identical conditions, indicative of their ignorable difference in structure and composition (Supplementary Fig. 25). The performances of enantiomeric mixtures were better than those of the diastereoisomeric mixtures, i.e., TpPa(Δ)-Cu(II)-COF/D-cysteine (10.48 mmol h⁻¹ g⁻¹) and TpPa(Λ)-Cu(II)-COF/L-cysteine (9.48 mmol h⁻¹ g⁻¹). Notably, the HER of TpPa(Δ)-Cu(II)-COF/L-cysteine was approximately 4-times higher than that of the achiral TpPa-Cu(II)-COF/L-cysteine (3.64 mmol h⁻¹ g⁻¹). The apparent quantum efficiency (AQE) at 600 nm drastically increased from 0.07% for TpPa-Cu(II)-COF/L-cysteine to 0.78% for TpPa(Δ)-Cu(II)-COF/L-cysteine (Supplementary Fig. 26).

A long-term photocatalytic activity was studied using the best performed catalytic system. No significant attenuation was observed in photocatalytic H₂ production under the 24-h irradiation for the TpPa(Δ)-Cu(II)-COF/L-cysteine (Supplementary Fig. 27). After every cycle, the collected photocatalysts were rinsed with 1 M HCl to eliminate cystine as it precipitated out and mixed with the COF solid (Supplementary Fig. 28). Although the acid treatment caused the coordination decomposition between TpPa(Δ)-COF and Cu(II), the obtained TpPa-COF solid could be re-coordinated with the equal quantity of Cu(II) for the following photocatalytic cycle. For the recycled TpPa(Δ)-COF, a series of characterizations including PXRD, N₂ sorption, FT IR, UV–vis and CD spectra corroborated that no remarkable changes were observed on compositions, structures and photophysical properties (Supplementary Fig. 29).

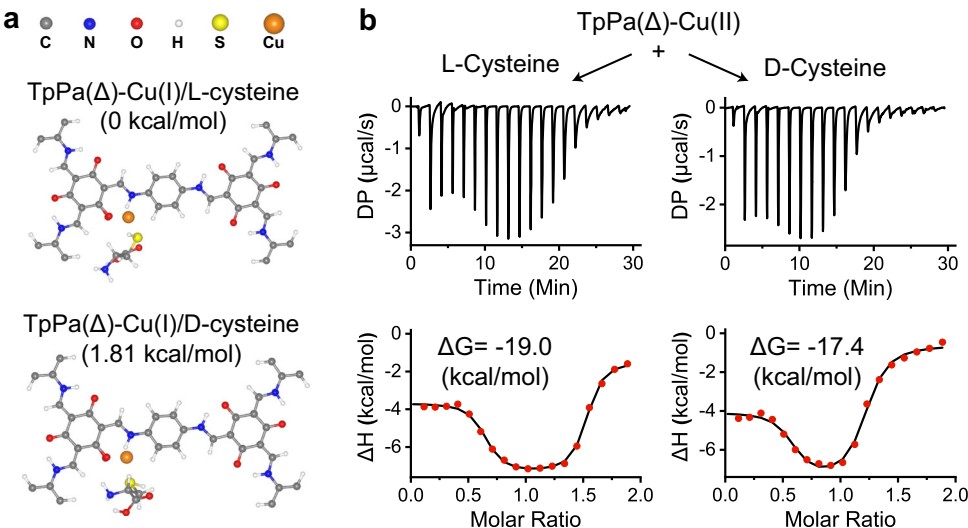

**Fig. 4 | Binding affinity between chiral COF and cysteine. a** Calculation of relative Gibbs free energy for the binding of TpPa(Δ)-Cu(I)-COF model with L-/D-cysteine, respectively. **b** ITC thermogram resulting from titration of a TpPa(Δ)-Cu(II)-COF dispersion (2 mmol/L) with a L-/D-cysteine aqueous solution (20 mmol/L) (up) and fitting with two sets of sites model (bottom). Source data are provided as a Source Data file.

To rationalize the origin of enantioselective photocatalysis, we examined the photophysical properties of the chiral and achiral TpPa-Cu(II)-COFs. They offered similar transient photocurrent responses (~$10^{-7}$ A cm$^{-2}$), resistances, and excited-state lifetimes (~0.28 ns) (Supplementary Figs. 30–32). Also, it was confirmed by XPS and EXAFS that the atomically dispersed Cu(II) ions were converted into Cu(I) within the chiral COF skeleton for hole extraction and electron transfer (Supplementary Figs. 14, 33, 34 and Supplementary Table 9). These findings elucidate the similar photophysical properties and copper valence change for the chiral and achiral TpPa-Cu(II)-COF in photocatalysis.

The interaction between chiral COF and cysteine enantiomer was estimated by the theoretical calculation for quantification[55]. The calculated Gibbs free energy for the binding of the TpPa(Δ)-Cu model with L-cysteine was 1.81 kcal mol$^{-1}$ lower than that for the TpPa(Δ)-Cu model/D-cysteine (Fig. 4a). Analogously, the binding of TpPa(Λ)-Cu model with D-cysteine conferred the relatively low Gibbs free energy of 1.77 kcal mol$^{-1}$ (Supplementary Fig. 35). By using the isothermal titration calorimetry, the ΔG values were obtained to prove that the binding energy of TpPa(Δ)-Cu(II)-COF with L-cysteine was relatively higher than that with D-cysteine (Fig. 4b). The enantioselective combination is energetically favorable to generating an enormous number of enantiomeric mixtures, i.e., TpPa(Δ)-Cu(II)-COF/L-cysteine and TpPa(Λ)-Cu(II)-COF/D-cysteine, thereby significantly improving the oxidation reaction kinetics.

The photocatalytic performances of enantiomeric mixtures were studied under different pH conditions. The optimized HER (14.72 mmol h$^{-1}$ g$^{-1}$) was achieved for the TpPa(Δ)-Cu(II)-COF(10.39 wt %)/L-cysteine at pH = 5.2, which is the isoelectric point of L-cysteine (Supplementary Fig. 36). When the pH values deviated from pH = 5.2 in the range of pH = 1–11, the HERs were significantly declined to be less than 2 mmol h$^{-1}$ g$^{-1}$. The result well corroborates that the neutral cysteine in the isoelectric state is more favorably assembled with the chiral COFs into the enantiomeric mixtures, leading to an increase in the SED oxidation kinetics.

Then, we focused on the proton reduction activity of TpPa(Δ)-COF evaluated by the Tafel-polarization curves. As shown in Fig. 3f, the y-intercept for TpPa(Δ)-COF is lower, unraveling that the chiral COF skeleton bears the smaller overpotential at the same current density to expedite the proton reduction kinetics. The computational studies were carried out to evaluate the electrostatic potential (ESP) of the cutout models from TpPa-Cu(I)-COF (Fig. 5a). In the excited state, the

electron depletion appeared surrounding the Cu(I) ion, which caused difficulty for the adsorption and reduction of positively charged protons. Comparably, the Tp moiety enriched electrons to render the strong affinity towards proton. As demonstrated above, the photo-generated electrons migrated from the Tp-Cu(I) complex to Tp moiety via the phenyl linker. Thus, we inferred that the catalytic sites of proton reduction should be located at the electron-accepting Tp moiety so that the binding free energy between Tp and H atom was calculated. As the H atom was put on the oxygen of C=O, the binding free energy was the lowest compared with the others on the Tp moiety (Fig. 5b), manifesting that the coupling of H atoms could occur at the keto sites.

The energy barriers of H$_2$ evolution were further evaluated for the chiral and achiral TpPa-Cu(II)-COFs at the given sites. The reaction undergoes the adsorption, reduction, and coupling of two protons on the COF skeletons by following the Volmer-Tafel mechanism. The rate-limiting step is the coupling of two H atoms in the transition state (TS). The high energy barriers of TS may arise from the isolation of adsorbed H atoms on the two neighboring layers, which impedes H$_2$ production. To reduce the energy barriers, the alignment of stacked layers is assumed to play an essential role, as it decides the spatial positions of adsorbed H atoms. By calculating the Gibbs free energy of the reaction pathway (Fig. 5c), the chiral TpPa-COF conferred a much lower energy barrier than the achiral TpPa-COF. It is primarily due to the difference in the stacking sequence of adjacent layers. As is known from the CD results, the chiral TpPa-COF allows for the parallel stacking of the adjacent layers, of which each atom of the upper layer is superimposed onto that of the lower layer. On the contrary, the achiral TpPa-COF prefers to stack layers antiparallelly, as indicated by the optimized layered structures (Supplementary Fig. 37). When the two adsorbed H atoms are coupled into H$_2$, the nearer distance between them is the more kinetically favorable. As displayed in Fig. 5d, the parallel alignment of the chiral TpPa-COF layers makes the two bonded H atoms more approach. As a result, the covalent connection of two separate H atoms from the adjacent layers is kinetically intensified in the chiral TpPa-COF.

## Discussion

With all the findings in mind, we compiled the HER data of the different photocatalytic systems in Table 1. Without any metal co-catalysts, the achiral and chiral TpPa-COF merely offer the lowest HER of 0.04 and 0.09 mmol g$^{-1}$ h$^{-1}$ in the presence of cysteine (Entries 1, 2). To

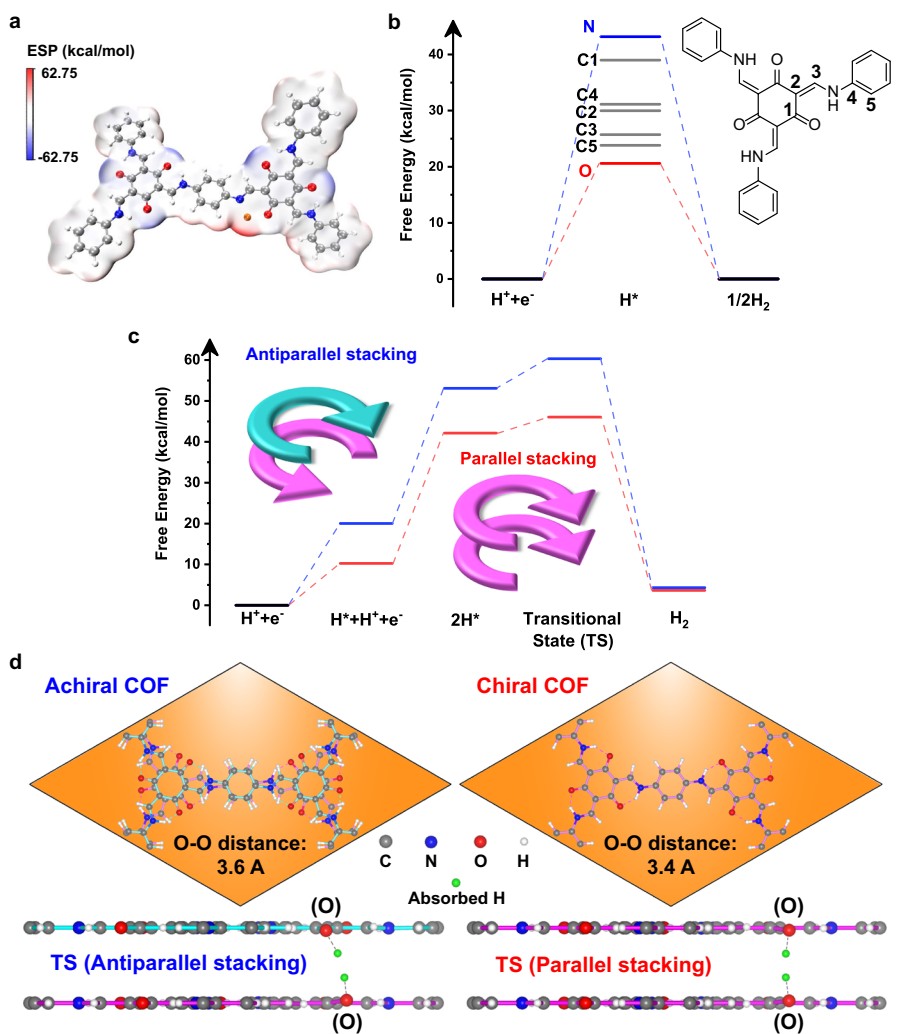

**Fig. 5 | Proposed kinetic mechanism of H2 production on the TpPa-Cu-COF.** **a** Electrostatic potential (ESP) of the TpPa-Cu(I) model. **b** Calculated binding free energy of H atom on the TpPa model. **c** Schematic Gibbs free energy diagrams for the H₂ evolution pathway on the two COF models. **d** Top-view and side-view of the parallelly stacked chiral TpPa-COF and antiparallelly stacked achiral TpPa-COF lattice models. Source data are provided as a Source Data file.

**Table 1 | Comparison of hydrogen evolution rate (HER) and sacrificial oxidation turnover frequency (TOF_ox) for the different combinations of COFs and cysteine**

| Entry | Sample | Cu(II) (wt%)[a] | Cysteine | HER (mmol h⁻¹ g⁻¹)[c] | TOF_ox (h⁻¹)[d] |
|---|---|---|---|---|---|
| 1 | TpPa-COF | None | L | 0.04 | – |
| 2 | TpPa(Δ)-COF | None | L | 0.09 | – |
| 3 | TpPa-COF | 4.26 mg[b] | L | 0.53 | 0.3 |
| 4 | TpPa-Cu(II)-COF | 4.99 | L | 0.40 | 0.5 |
| 5 | TpPa-Cu(II)-COF | 8.47 | L | 2.30 | 1.7 |
| 6 | TpPa-Cu(II)-COF | 10.76 | L | 3.64 | 2.2 |
| 7 | TpPa-Cu(II)-COF | 12.77 | L | 2.94 | 1.5 |
| 8 | TpPa(Λ)-Cu(II)-COF | 9.72 | L | 9.48 | 6.3 |
| 9 | TpPa(Λ)-Cu(II)-COF | 9.72 | D | 12.80 | 8.5 |
| 10 | TpPa(Δ)-Cu(II)-COF | 10.51 | L | 14.72 | 9.0 |
| 11 | TpPa(Δ)-Cu(II)-COF | 10.51 | D | 10.48 | 6.4 |

[a]The weight percentage of coordinated $Cu^{2+}$ ions in the TpPa-Cu(II)-COF solid was determined by ICP-AES.
[b]$Cu(OAc)_2 \cdot H_2O$ (4.26 mg) was dissolved in the aqueous dispersion (100 mL) of TpPa-COF (10 mg). The content of $Cu^{2+}$ ions relative to the COF solid is 10.8 wt%, similar to Entry 6.
[c]All the HERs measured using the same instruments, optical setup, and reaction conditions: 10 mg COF solid, 0.1 M cysteine, 100 mL water, 300 W Xe light source equipped with λ > 420 nm cut-off filter (200 mW cm⁻²). HERs were obtained from the 6-h photoirradiation and normalized to the sample mass.
[d]All the TOF_ox values were calculated by the total amount of Cu in the oxidation reaction.

ameliorate the kinetics, a redox couple of Cu(II) and cysteine is applied to accelerate the photo-induced hole extraction with the electron mediator Cu(I), which is formed by the dark redox reaction between cysteine and Cu(II). The mixture of achiral TpPa-COF and redox couple allows the dramatic increase in the HER ($0.53 \, mmol \, g^{-1} \, h^{-1}$, Entry 3), while the coordinated Cu(II) within TpPa-COF (Entries 4–7) plays the more positive role in the $H_2$ evolution under identical conditions ($3.64 \, mmol \, g^{-1} \, h^{-1}$, Entry 6). It is due to the further enhancement in the kinetics of hole extraction by the reaction of atomically dispersed Cu(II) with cysteine. Nevertheless, continuously raising the number of coordinated Cu(II) is unfavorable to the HER ($2.94 \, mmol \, g^{-1} \, h^{-1}$, Entry 7), as the metal coordination with N-salicylideneaniline consumes many H-adsorbed sites on the C=O groups. Subsequently, the enantioselective photocatalytic system is skillfully established by combining the TpPa(Δ)-Cu(II)-COF with L-cysteine or TpPa(Λ)-Cu(II)-COF with D-cysteine. Their corresponding HERs are as high as 14.72 (Entry 10) and $12.80 \, mmol \, g^{-1} \, h^{-1}$ (Entry 9), respectively, which are 3-4-fold higher than that of the achiral TpPa-Cu(II)-COF/L-cysteine and approximately 1.5-times higher than those of the diastereomeric mixtures (Entries 8,11) under identical conditions. Without the coordinated Cu(II), the same outcome on the photocatalytic behavior was observed for the chiral TpPa-COF with its cysteine enantiomer (Supplementary Fig. 38). It can be attributed to the intensified oxidation reaction kinetics by docking cysteine onto the chiral COFs via the enantioselective combination. We found that the strategy applied to the other chiral SEDs, such as L-ascorbic acid or D-araboascorbic acid (Supplementary Fig. 39). Also, when Pt nanoparticles (~2 nm) were photo-deposited onto the chiral TpPa-COF (Supplementary Fig. 40), the HERs were promoted through the enantioselective combination of the chiral TpPa-COF/Pt with SEDs, outperforming the diastereomeric mixtures (Supplementary Fig. 41) and the achiral TpPa-COF/Pt ($8.42 \, mmol \, g^{-1} \, h^{-1}$)[39]. The findings underpin the generality of our strategy without relying on specific metals and SEDs.

To conclude, we have proposed an unprecedented strategy to explore the potential of the chiral 2D COF for photocatalytic $H_2$ evolution in water. The well-known β-ketoenamine-linked 2D COFs are constructed using the aldimine reaction of Tp with diamines, followed by the in-situ enol-to-keto transformation. A high concentration of atomically dispersed Cu(II) ions is coordinated with N-salicylideneaniline at the nodes of hexagonal frameworks, serving as electron transfer mediators for SED oxidation and hole extraction via a circular conversion between Cu(II) and Cu(I). Consequently, the enantioselective combination of the chiral TpPa-Cu(II)-COF with L-/D-cysteine as the SED is exposed to the visible irradiation in water (pH = 5.2) for $H_2$ evolution. The record high HER of $14.72 \, mmol \, g^{-1} \, h^{-1}$ is achieved for the enantiomers of TpPa(Δ)-Cu(II)-COF/L-cysteine without added precious metal co-catalyst. Such a performance remarkably outperforms those of the achiral TpPa-Cu(II)-COF with L-cysteine and the achiral TpPa-COF using Pt nanoparticles as co-catalyst. The distinct superiority can be attributed to the enhanced reaction kinetics for cysteine oxidation and proton reduction. As a key driving force, the chiral 2D COF affords strong enantiomeric interaction to dock SEDs and is favorable for the parallelly superimposed stacking of layers, contributing to the reduced energy barriers of $H_2$ production. Therefore, our study is pioneered for the structure-to-activity correlation of the chiral 2D COFs in photocatalytic $H_2$ evolution and would stimulate the exploration of non-precious-metal organic photocatalysts.

## Methods

### Synthesis of chiral TpPa-Cu(II)-COF

A small vial was charged with Tp (16 mg, 0.075 mmol), (S)- or (R)-1-benzenemethanamine (9 mg, 0.075 mmol), and a mixture of mesitylene and dioxane (1.5 mL, 1:1 v/v). The mixture was sonicated for 5 min to give an orange solution, followed by the addition of p-phenylenediamine (Pa) (12.1 mg, 0.112 mmol). The obtained dispersion was

transferred into a Pyrex tube. Then aqueous acetic acid (0.3 mL, 6 M) was added to the mixture as a catalyst. The tube was degassed by three freeze–pump–thaw cycles and sealed off. Then the reaction proceeded in an oven at 120 °C for 3 days. The precipitate was filtered off, washed with THF (3 × 10 mL), extracted by Soxhlet with THF for 24 h, and dried at 40 °C under vacuum overnight to give the product with a yield of 66–72%. Next, 20 mg chiral TpPa-COF was dispersed in the aqueous solution of copper acetate (2 mL, 0.2 M). The suspension was stirred at 150 rpm and incubated at 50 °C for 12 h. The resultant product was filtered off, washed with deionized water (5×6 mL) and THF (3 × 6 mL), and dried at 40°C under vacuum to obtain the chiral TpPa-Cu(II)-COF.

### Photocatalytic $H_2$ evolution

The photocatalyst (10 mg) and L-/D-cysteine (1.21 g, 0.1 M) were dispersed in deionized water (100 mL) under sonication, and the mixture was charged in a top-irradiation quartz photoreactor connected with the labsolar 6A system (Beijing Perfect Light Technology Co., Ltd). Before the reaction, the aqueous dispersion was purged with Ar for 30 min and exposed to a vacuum for 20 min. The reaction was kept at 10 °C with circulating cooling water, stirred at a constant rate, and irradiated by a 300-W Xe lamp (Perfect Light PLS-SXE300UV) equipped with a cut-off filter (>420 nm). The irradiation power of the light-spot center was adjusted to $200 \, mW \, cm^{-2}$ calibrated by the optical power meter (Aulight GEL-NP2000). $H_2$ product was determined by the online gas chromatography (Techcomp GC7900). The HERs were obtained from a linear regression fit. For recycling, the photocatalyst was recovered after washing with 1 M HCl, drying, and then complexing with $Cu^{2+}$ at 50 °C for 12 h.

## Data availability

All data supporting the findings of this study are available with the article, as well as the Supplementary Information file, or available from the corresponding authors upon reasonable request. Source data are provided with this paper.

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

## Acknowledgements

This work is supported by the National Natural Science Foundation of China (Grants Nos. 51973039 and 52173197).

## Author contributions

J.G. conceived and supervised the research. W.W. synthesized the catalysts, conducted photocatalytic measurements, and performed DFT calculations. J.G. and W.W. analyzed the data and wrote the paper.

## Competing interests

The authors declare no competing interests.
