## [Peer Review File · Nature Communications]

Title: The Effect of Enantioselective Chiral Covalent Organic Frameworks and Cysteine Sacrificial Donors on Photocatalytic Hydrogen EvolutionREVIEWER COMMENTS

Reviewer #1 (Remarks to the Author):

Enantioselective Combination of Chiral Covalent Organic Frameworks with Cysteine Sacrificial Donors for Enhanced Reaction Kinetics in Photocatalytic Hydrogen Evolution by Weijun Weng and Jia Guo

Weng and Guo reports on unique design of a chiral COF system as an efficient materials platforms for photocatalytic reactions with unique Cu(II) centers. The outcomes from the materials, especially the design of COFs, are all excellent and surprising for all working on conjugated COFs as electronic functional materials, thus in principle I have to say words of praise for the idea. The center of the unique idea is: 1) chiral COF photocatalysts and complexed with Cu(II), 2) the effect of chirality of sacrificial electron donor (SED) over the photocatalytic H₂ evolution, and 3) the chiral recognition for the docking of SED in the Cu(II) center for efficient hole quenching due to the close proximity. The last strategy is conceptually excellent, and supported well by various experimental results.

In contrast, the most critical issue is in the photocatalytic H₂ evolution values because COF synthesized in different batches exhibit slightly different catalytic activity due to the difference in crystallinity, porosity or packing. Also, I think most of the sacrificial electron donors will be in dynamics and may not dock on the chiral Cu(II) center during the reaction. The latter is in particular often leading significant differences in the catalytic activity.

Herein the authors have put forward this assumption, "Meanwhile, the chiral crystalline frameworks favorably form the parallelly superimposed stacking of layers, which remarkably lowers the energy barriers of H-H coupling at the keto sites around the chiral center." This claim is, unfortunately supported by no experimental proofs so far.

One another issue is in the role of Cu in the catalytic system. The authors have mentioned that Cu acts as an electron mediator and the Tp unit as the catalytic center for proton reduction. I am doubtful about this assumption, because previously several Tp based COFs are reported as the photocatalyst but they exhibit catalytic activity only in the presence of a metal co-catalyst (generally Pt). It is mandatory to address the points with concrete base of experimental evidences.

To high rise of the present work for the future design of COF-based catalytic systems, I would like to recommend the following discrete set of additional experiments:

- 1) Check carefully the photocatalytic activity experiment under different pH conditions and compare all the data.
- 2) Identify the fate of sacrificial electron donor molecules, in this case L-cysteines. With my understandings, the reaction schema of cysteine under oxidation reaction are presumed to shift abruptly against pK_a condition, and we will be able to trace the route by checking the products from SEDs. If the reaction route is as expected of mine, the resulted product molecular analysis will give an important information on reactive intermediates against electron donation to COF frames. I am presuming different scenario on acceleration of H₂ propagation reactions from the authors' a bit

obscure claims in the present version of the manuscript. If in the case, impact of the present work will be jumped up, shocking all working in the related field.

3) Also, the authors are requested to perform the photocatalysis using L- and D-ascorbic acid and compare with the present result, that will provide detail insight for the chiral docking. Also, the authors can different chiral and achiral SED for the photocatalysis.

4) The photocatalytic activity of the pristine chiral COFs without Cu should be evaluated using Pt as co-catalyst and different chiral SED. This result will help the authors to validate their hypothesis.

Reviewer #2 (Remarks to the Author):

In this manuscript, the authors provide a strategy of the enantioselective combination of the chiral TpPa-Cu(II)-COF with L-/D-cysteine as the SED for photocatalytic H₂ evolution. The catalyst design is novel and the results and discussion are sufficient and deep. In my opinion, it could be accepted after major revision.

1) It is unclear for the synthesis of the TpPa-Cu(II) model in the Supplementary Information. What is the accurate reaction mechanism for the keto-to-enol transformation? Why is it not the irreversible enol-to-keto transformation? What are the key parameters for determining the final products?

2) The authors highlighted the single-atom Cu (II) distribution in TpPa-COF. How to identify the valence state of single-atom Cu except for XPS spectra? It is generally considered that the metal single atom is presented as M (0) anchored on COFs in the references (Chem. Eng. J., 403 (2021) 126383; ACS Catal., 11 (2021) 13266-13279; J. Am. Chem. Soc., 141 (2019) 7615-7621.) What accounts for the difference?

3) From Fig. 1c, the authors mentioned that the other emerging peak at 398.5 eV was ascribed to the newly formed C=N bond. Thus, the authors suggested the occurrence of the keto-to-enol transformation. It is an important conclusion for the proposed structure of the TpPa-Cu(II)-COF complex. However, why the C=N bond is absent in the FTIR spectra?

4) Some related typical references on the TpPa-COF for photocatalytic hydrogen evolution should be cited in the Introduction section and compared in Table S8, such as Angew. Chem. 2018, 130, 1-6; ACS Catal., 11 (2021) 13266-13279.

5) In Fig. 2a, why the potential of H⁺/H₂ is presented at pH=5.5? What is the exact potential for cysteine oxidation at the same pH value?

6) In Fig. S15, the signal for the mixture of Cu(II) ions and L-cysteine after 5 min is so weak that it can not be evidence to support the DMPO-trapped sulfanyl radicals. Maybe the authors could extend the irradiation time to detect the signal.

7) In-Page 11, the authors mentioned that electrons transfer from the Cu(I) donor to the photogenerated acceptor on the excited TpPa-COF upon visible irradiation. Why is it not from the excited TpPa-COF to the Cu(I)? How to confirm the electron's transfer direction?

8) In Fig. S26, the authors completed the photocatalytic recycling performances of TpPa(Δ)-Cu(II)-COF under 24-h irradiation of visible light. They mentioned that the recycled sample was washed with 1M HCl to remove the remaining Cu(II) ions after one cycle of 6-h photocatalysis, and loaded with similar

content of fresh Cu(II) ions for the next photocatalytic cycle. Does it mean the Cu(II) ions are easily removed during the photocatalytic process for the sample of TpPa(Δ)-Cu(II)-COF? If that, the Cu(II) ions in TpPa(Δ)-Cu(II)-COF is not stable. The authors should explain it in detail.

Reviewer #3 (Remarks to the Author):

The manuscript report the enantioselective combination of a chiral ketoenamine-linked COF with L-/D-cysteine as the sacrificial electron donor for hydrogen evolution. Without precious 22 metal co-catalysts, the photocatalytic enantiomers can significantly enhance the H₂ evolution 23 rate of up to 14.72 mmol h⁻¹g⁻¹ with a high sacrificial oxidation turnover frequency of 9.0 h⁻¹. The origin of superior performance lies in the increase in the reaction kinetics. There some problems need be addressed.

1. The chiral induced spin selectivity (CISS) effect has been well studied in nanostructured inorganic and polymeric materials (eg Acc. Chem. Res. 2020, 53, 2659–2667).Introducing chirality into metal-semiconductor hybrid nanostructures can boost the chiral hot electrons or spin-selective electrons of plasmonic nanocomponents, which can transfer to a catalytic semiconductor and trigger asymmetric catalytic reactions.Very recently, Wang et al reproetd that the chiral hybrid nanostructures can drive chirality-dependent photocatalytic hydrogen generation (doi.org/10.1002/anie.202112400).Unfortunately, all these previous reports are not mentioned in the manuscript.

2.Strangely, the structure annd PHE of the achiral COF with copper salt were well studied and discussed, but those of the chral COF were even not provided. It is difficult to follow the main topic of thhis manuscript.

So, although this work is interesting, i cannot recommendate its publication.

REVIEWER COMMENTS

Reviewer #1 (Remarks to the Author):

Enantioselective Combination of Chiral Covalent Organic Frameworks with Cysteine Sacrificial Donors for Enhanced Reaction Kinetics in Photocatalytic Hydrogen Evolution by Weijun Weng and Jia Guo

Weng and Guo reports on unique design of a chiral COF system as an efficient materials platforms for photocatalytic reactions with unique Cu(II) centers. The outcomes from the materials, especially the design of COFs, are all excellent and surprising for all working on conjugated COFs as electronic functional materials, thus in principle I have to say words of praise for the idea. The center of the unique idea is: 1) chiral COF photocatalysts and complexed with Cu(II), 2) the effect of chirality of sacrificial electron donor (SED) over the photocatalytic H₂ evolution, and 3) the chiral recognition for the docking of SED in the Cu(II) center for efficient hole quenching due to the close proximity.

The last strategy is conceptually excellent, and supported well by various experimental results.

Response: We highly appreciate the affirmation of the reviewer to our work.

In contrast, the most critical issue is in the photocatalytic H₂ evolution values because COF synthesized in different batches exhibit slightly different catalytic activity due to the difference in crystallinity, porosity or packing.

Response: Thanks for the reviewer's suggestion.

We agree that the COFs synthesized in different batches may exhibit different PHE activity. Thus, the repeated photocatalytic tests with the different COFs were carried out, and the observed H₂ evolution performances were maintained.

The supplemented data is put in the *Supplementary Information*, **Supplementary Fig. 25**, and the revision has been made in the main text, *Page 15, Line 17-20*, "To validate the reliability of photocatalytic performances, the chiral TpPa(Δ)-Cu(II)-COFs were synthesized in the different batches, exhibiting the similar HERs under identical conditions, indicative of their ignorable difference in structure and composition (**Supplementary Fig. 25**)."

Supplementary Fig. 25 Time-dependent photocatalytic H₂ evolution using the various mixtures of TpPa(Δ)-Cu(II)-COF, TpPa(Λ)-Cu(II)-COF and TpPa-Cu(II)-COF with chiral cysteine, respectively. All the COFs used for photocatalysis were synthesized in the different batches (A, B and C) under identical conditions.

Also, I think most of the sacrificial electron donors will be in dynamics and may not dock on the chiral Cu(II) center during the reaction. The latter is in particular often leading significant differences in the catalytic activity. Herein the authors have put forward this assumption, "Meanwhile, the chiral crystalline frameworks favorably form the parallelly superimposed stacking of layers, which remarkably lowers the energy barriers of H-H coupling at the keto sites around the chiral center." This claim is, unfortunately supported by no experimental proofs so far.

Response: Thanks for the constructive suggestion of Reviewer #1.

We agree that the assumption "Meanwhile, the chiral crystalline frameworks..." lacks substantial experimental proofs. In the revised manuscript, the Tafel-polarization curves revealed that the chiral TpPa(Δ)-COF possessed a smaller overpotential than the achiral TpPa-COF (*Fig. 3f*), corroborating that the chiral crystalline framework bears the higher catalytic activity. Also, we supplemented the photocatalytic tests for the achiral TpPa-COF and chiral TpPa(Δ)-COF in the presence of D-cysteine or D-arabascorbic acid as the SED, respectively. Both systems could not dock the SED onto the TpPa-COFs. However, the diastereomeric mixture of chiral TpPa(Δ)-COF/D-SED offered higher H₂ evolution rates than the achiral TpPa-COF/D-SED (*Supplementary Fig. 38b*). Therefore, we correlated the photocatalytic activity with the parallelly superimposed stacking structure of chiral frameworks, predicting the catalytic mechanism of active sites.

According to the reviewer's suggestion, we revised this assumption, *Page 5, Line 4-7*, "Meanwhile, the chiral crystalline frameworks exhibit a higher catalytic activity and smaller overpotential in H₂ production than the achiral analogs, implying the lowered energy barriers of H₂ production for the chiral COFs."

The photocatalytic activity of the chiral TpPa-COF without the coordinated Cu(II) was assessed, and the related content has been supplied in the context, *Page 20, Line 9-11*, "Without the coordinated Cu(II), the same outcome on the photocatalytic behavior was observed for the chiral TpPa-COF with its cysteine enantiomer (*Supplementary Fig. 38*)."

Supplementary Fig. 38 Comparison of the HERs for the chiral and achiral TpPa-COF combined with various chiral SEDs, including L-cysteine (L-Cys), D-cysteine (D-Cys), L-ascorbic acid (L-AA), and D-araboascorbic acid (D-AA).

One another issue is in the role of Cu in the catalytic system. The authors have mentioned that Cu acts as an electron mediator and the Tp unit as the catalytic center for proton reduction. I am doubtful about this assumption, because previously several Tp based COFs are reported as the photocatalyst but they exhibit catalytic activity only in the presence of a metal co-catalyst (generally Pt). It is mandatory to address the points with concrete base of experimental evidences.

Response: Thanks for the reviewer's issue. The early reported photocatalytic system often adopts Pt nanoparticles as co-catalyst for hydrogen reduction. Compared with the Tp-based COF, Pt bears the deeper Fermi energy level and higher affinity to a proton, thereby providing the predominated catalytic sites for proton reduction and hydrogen coupling. When Pt co-catalysts do not exist, the photocatalytic sites are considered from the Tp-based COFs for H₂ evolution.

As addressed by the reviewer, the coordinated Cu(II) ions on the TpPa-COF are likely to be reductive sites. However, the standard potentials (vs. NHE, 25°C) of Cu²⁺/Cu⁺, Cu²⁺/Cu and

Cu⁺/Cu are 0.159V, 0.340V, and 0.520V, respectively, reflecting that various Cu ions are inert on proton reduction in the dark reaction. With light irradiation, Cu(0) has been reported to act as a co-catalyst to manage the photogenerated electrons for the reduction of adsorbed protons (*Applied Catalysis B: Environmental*, 2021, **284**, 119743), while only the Cu(I) or Cu(II) ions were detected in the TpPa-COF. Compared with Cu(0) metal, the high ESP around the Cu ions suppressed the adsorption and subsequent reduction of protons. It is inferred that the photoinduced H₂ evolution did not occur on the coordinated Cu ions. In the supplemented experiments, we found that the chiral TpPa-COF without Cu(II) or Pt(0) could catalyze the generation of H₂ in the presence of cysteine under visible irradiation, substantiating our assumption that the reductive sites are localized on the COF skeletons (*Supplementary Fig. 38*).

In addition, we supplemented the cyclic voltammogram (CV) measurement for the TpPa-Cu(II)-COF in the presence of L-cysteine (*Supplementary Fig. 16*). The cathodic current density can assess the H₂ production level in the electrochemical reaction. Given that the theoretical proton reduction potential is -0.51V (vs. Ag/AgCl), the cathodic current density at the more negative potential reveals the proton reducing ability of materials. TpPa-Cu(II)-COF and TpPa-COF gave a similar current density at -1.0 ~ -0.6V, disclosing that the complexed Cu(II) had no strengthened effect on the hydrogen production. For the Pt-deposited TpPa-COF, a significant increase in the cathodic current density was observed, implying that the loaded Pt worked as a co-catalyst to boost H₂ production under identical electrochemical conditions.

Apart from the current density, we investigated the onset oxidative potential of L-cysteine in the CV curves. When Cu²⁺ was added to the solution, the onset potential dramatically decreased from 0.79V to 0.21V. Also, it was observed that the TpPa-Cu(II) COF offered the lower onset potential of L-cysteine at 0.17V. In the enantioselective mixture of TpPa-Cu(II) and L-cysteine, the potential was kept at 0.16V. Without Cu²⁺ coordinated in the TpPa-COF, the oxidative potentials of L-cysteine were restored at around 0.5V. All the results elucidate that the coordinated Cu²⁺ plays a crucial role in the catalytic oxidation of cysteine.

Therefore, taken all together, we believe that the coordinated Cu(II) predominantly takes part in the SED oxidation. Furthermore, based on the theoretical analysis, the formed Cu(I) acts as the mediator for electron transfer towards the catalytic sites on the COF (*Please see the details in the Q7 of Reviewer 2#'s report, Page R20*).

Supplementary Fig. 16 Cyclic voltammogram curves of various samples, using the N_2 -bubbled $0.2\text{M Na}_2\text{SO}_4$ aqueous solution as the electrolyte at the scan rate of 100 mV s^{-1} . The concentration of L-Cysteine is 0.1 mol/L ($\text{pH}=5.2$). The concentration of $\text{Cu}(\text{OAc})_2$ is 0.2 mmol/L .

The supplemented CV tests and discussions have been added in the context, *Page 11, Line 17-22*, and *Page 12, Line 1-8*, “The observed oxidative potential of L-cysteine (0.1 M) was 0.79V (vs. Ag/AgCl) in the N₂-saturated solution at pH 5.2. With the same conditions, the onset potential of L-cysteine was largely shifted to 0.21V (vs. Ag/AgCl) in the presence of Cu²⁺ ions (2×10⁻⁴ M). In contrast, when the TpPa-Cu(II)-COF solid was coated on the electrode, the oxidative potential could continuously decrease to 0.17V (vs. Ag/AgCl). It discloses that the TpPa-COF component strengthens the coordinated Cu(II) electrophilicity to oxidize L-cysteine. On the other hand, the cathodic current density of the TpPa-Cu(II)-COF at -1.0 ~ -0.6V (vs. Ag/AgCl) was similar to that of the TpPa-COF. Nevertheless, both were far lower than that of the Pt-deposited TpPa-COF. As the theoretical proton reduction potential is -0.51V (vs. Ag/AgCl), the cathodic current density at the more negative potentials reflects the catalytic ability of materials in the electrochemical reaction. It is therefore likely that the catalytic activity of the complexed Cu(II) ions for H₂ production is ignorable, while its role in the L-cysteine oxidation is predominated.” The CV figure was added in the *Supplementary Information, Supplementary Fig. 16*.

To high rise of the present work for the future design of COF-based catalytic systems, I would like to recommend the following discrete set of additional experiments:

1) Check carefully the photocatalytic activity experiment under different pH conditions and compare all the data.

Response: According to the reviewer’s suggestion, the photocatalytic activity has been carefully examined under different pH conditions. As shown in *Supplementary Fig. 36*, the optimum HER is achieved for the TpPa(Δ)-Cu(II)-COF(10.39wt%)/L-cysteine at pH 5.2, which is the isoelectric point of L-cysteine. When the pH values deviated from 5.2, the HERs dramatically declined. We reason that the positively or negatively charged L-cysteine was well dissolved in an aqueous solution, severely impairing the formation of enantiomeric mixtures. Thus, as the reviewer addressed, most SEDs are in dynamics and cannot be docked onto the chiral COF frameworks to improve the photocatalytic reaction kinetics.

The change has been made in the context, *Page 17, Line 14-21*, “The photocatalytic performances of enantiomeric mixtures were studied under different pH conditions. The optimized HER (14.72 mmol h⁻¹g⁻¹) was achieved for the TpPa(Δ)-Cu(II)-COF(10.39wt%)/L-cysteine at pH = 5.2, which is the isoelectric point of L-cysteine (*Supplementary Fig. 36*). When the pH values deviated from pH = 5.2 in the range of pH = 1~11, the HERs were significantly declined to be less than 2 mmol h⁻¹g⁻¹. The result well corroborates that the neutral cysteine in the isoelectric state is more favorably assembled with the chiral COFs into the enantiomeric mixtures, leading to an increase in the SED oxidation kinetics.” The related figure is added in the *Supplementary Information, Supplementary Fig. 36*.

Supplementary Fig. 36 Photocatalytic H₂ evolution of TpPa(Δ)-Cu(II)-COF(10.39wt%) with L-cysteine under different pH values. The pH of the solution was unchanged during the photocatalytic process.

2) Identify the fate of sacrificial electron donor molecules, in this case L-cysteines. With my understandings, the reaction schema of cysteine under oxidation reaction are presumed to shift abruptly against pKa condition, and we will be able to trace the route by checking the products from SEDs. If the reaction route is as expected of mine, the resulted product molecular analysis will give an important information on reactive intermediates against electron donation to COF frames. I am presuming different scenario on acceleration of H₂ propagation reactions from the authors' a bit obscure claims in the present version of the manuscript. If in the case, impact of the present work will be jumped up, shocking all working in the related field.

Response: We appreciate the reviewer for suggesting an insightful study on the enantioselective photocatalytic system.

We have evaluated the HERs under different pH conditions. As cysteine is in the isoelectric state at pH = 5.2, its enantioselective coupling with the chiral COF is the most favorable compared to the other forms of charged cysteine. Thus, donating electrons from the docked cysteine to COF skeletons is of high efficiency. As the pH values deviated from its pI, the charged cysteine had good solubility in an aqueous solution, and in turn, the enantioselective assembly between chiral COF and L-/D-cysteine was remarkably impaired.

Figure R1. ^1H NMR spectra of L-cysteine solution before (bottom) and after (up) photocatalysis under different pH conditions.

According to the reviewer, we identified the product species in the oxidation of cysteine by the ^1H NMR measurement. The spectra revealed that all the products were cystine formed by the disulfide coupling of cysteine (**Figure R1**). The reactive intermediate is assumed to be the radical anions ($\text{CysS}^{\bullet-}$), confirmed by the EPR signals (**Supplementary Fig. 15**).

3) Also, the authors are requested to perform the photocatalysis using L- and D-ascorbic acid and compare with the present result, that will provide detail insight for the chiral docking. Also, the authors can different chiral and achiral SED for the photocatalysis.

Response: Thanks for the valuable suggestion of the reviewer.

We performed the photocatalysis using L- ascorbic acid (L-AA) and D-araboascorbic acid (D-AA) as the SEDs with the chiral TpPa-Cu(II) COFs. As shown in **Supplementary Fig. 39**, the enantioselective combinations between COFs and AA, *i.e.*, TpPa(Δ)-Cu COF/L-AA and TpPa(Λ)-Cu COF/D-AA, offer the higher H_2 evolution rates and the more significant turnover frequencies for SED oxidation than those of the diastereomeric mixtures. These results unravel that the enantiomeric docking plays a crucial role in the increase in reaction kinetics of photocatalysis.

The change has been made in the main text, *Page 20, Line 11-14*, “It can be attributed to the intensified oxidation reaction kinetics by docking cysteine onto the chiral COFs via the enantioselective combination. We found that the strategy applied to the other chiral SEDs, such as L-ascorbic acid or D-araboascorbic acid (**Supplementary Fig. 39**).” The supplemented figures are put in the *Supplementary Information, Supplementary Fig. 39*.

Supplementary Fig. 39 Comparison of (a) the HERs and (b) TOFs for the TpPa(Δ)-Cu(II)-COF(10.39wt%), TpPa(Λ)-Cu(II)-COF(9.63wt%) combined with the chiral L-ascorbic acid (L-AA) and D-araboascorbic acid (D-AA), respectively. The pH of the solution was adjusted to be 5.2 with sodium L-ascorbate or sodium D-araboascorbate. The stars on the histograms represent the different enantioselective mixtures.

4) The photocatalytic activity of the pristine chiral COFs without Cu should be evaluated using Pt as co-catalyst and different chiral SED. This result will help the authors to validate their hypothesis.

Response: According to the reviewer’s suggestion, we performed the reaction using the pristine chiral TpPa-COF without the coordinated Cu(II). The photo-deposited Pt nanoparticles served as a co-catalyst and the chiral SEDs including ascorbic acid and cysteine were assembled with the chiral COFs into the enantiomeric mixtures, respectively. In the presence of Pt nanoparticles, the enantioselective combination facilitates the docking of SED onto the chiral skeletons of COFs, resulting in the higher H₂ evolution rates than that of the diastereomeric mixtures under the identical conditions. Therefore, regardless of Pt(0) or Cu²⁺ immobilized in the COFs, docking SEDs is essential for the SED oxidation kinetics and significantly enhances photocatalytic H₂ evolution performance.

A detailed discussion about the photocatalysis using Pt as co-catalyst and different chiral SEDs has been added in the main text, *Page 20, Line 14-19*, “Also, when Pt nanoparticles (~2 nm) were photo-deposited onto the chiral TpPa-COF (**Supplementary Fig. 40**), the HERs were promoted through the enantioselective combination of the chiral TpPa-COFs/Pt with SEDs, outperforming the diastereomeric mixtures (**Supplementary Fig. 41**) and the achiral TpPa-COF/Pt (8.42 mmol

$\text{g}^{-1}\text{h}^{-1})^{39}$. The findings underpin the generality of our strategy without relying on specific metals and SEDs.” The related figure is supplemented in the *Supplementary Information, Supplementary Fig. 40 and Fig.41*.

Supplementary Fig. 40 TEM images of (a,b) the TpPa(Δ)-COF/Pt and (c) the achiral TpPa-COF/Pt. The deposited Pt amount was $2.8 \pm 0.2\text{wt}\%$. Insets in (b) and (c) display the statistical size distributions of the photo-deposited Pt nanoparticles.

Supplementary Fig. 41 Comparison of the HERs and TOFs for the Pt-deposited chiral TpPa-COFs combined with (a) L-Cysteine (L-Cys) and D-Cysteine (D-Cys) and (b) L-ascorbic acid (L-AA) and D-araboascorbic acid (D-AA). The deposited Pt amount was 2.8 ± 0.2 wt% and the used chiral SEDs was 0.1M. The stars on the histograms represent the different enantioselective mixtures.

Reviewer #2 (Remarks to the Author):

In this manuscript, the authors provide a strategy of the enantioselective combination of the chiral TpPa-Cu(II)-COF with L-/D-cysteine as the SED for photocatalytic H₂ evolution. The catalyst design is novel and the results and discussion are sufficient and deep. In my opinion, it could be accepted after major revision.

1) It is unclear for the synthesis of the TpPa-Cu(II) model in the Supplementary Information. What is the accurate reaction mechanism for the keto-to-enol transformation? Why is it not the irreversible enol-to-keto transformation? What are the key parameters for determining the final products?

Response: Thanks for the reviewer's issue.

As early reported, the enol-to-keto transformation in the Tp model is thermodynamically driven to form the stable keto-enamine structure in the ground state (*J. Am. Chem. Soc.* 2013, **135**, 5328; *Phys. Chem. Chem. Phys.*, 2021, **23**, 1156). Such configuration is available to complex with metal (*Nat Commun.*, 2018, **9**, 1294.), while a detailed study on the complexation has been rarely reported. Here, we propose a reaction mechanism of metal-complexation-induced structural transformation.

The Mayer Bond Order (MBO) is usually applied to describe the bond strength and correlates with the empirical bond order (*J. Comput. Chem.* 2007, **28**, 204). The molecular models were optimized and calculated at the level of PBE0-D3BJ/def2-TZVP for estimation of the MBO. Compared with the Tp model, the MBO in the Tp-Cu(II) model is strengthened from 1.37 to 1.54 between N and C atoms and weakened from 1.72 to 1.13 between O and C atoms (**Figure R2**). The changes illustrate the rearrangement of electron density within the different bonds. It is accordingly inferred that the single bond of enamine (C-N) is transformed into the double bond of imine (C=N), and the double bond of keto (C=O) is transformed into the single bond of enol (C-OH). As a result, from the computational simulation, the Cu(II) complexation induces the electron rearrangement and, in turn, the keto-to-enol transformation.

The key parameters for determining the final product were derived from the FT IR and XPS spectra in **Supplementary Fig. 5** and **Fig. 1b**. When compared to the model complex Tp-Cu(II), the newly formed C=N bond in the Tp-Cu(II) structure was identified by the characteristic vibration band at 1601 cm⁻¹ in the FT IR spectrum. Also, the deconvolution of the N 1s core-level XPS peak revealed the presence of C=N bond (398.1 eV). As the coordinated quantity of Cu ions was ~10wt%, the signals of imine bonds in the TpPa-Cu(II) COF are medium in intensity, while the unequivocal XPS signal of C=N can be observed at 398.1 eV.

Figure R2. Mayer bond order of Tp model and Tp-Cu(II) model. N: blue, O: red, C: grey, H: white, and Cu: Orange.

The discussion regarding the mechanism is added in the main text, *Page 7, Line 7-9*, “We assume that the reversed tautomerization is caused by the metal complexation, which leads to the electron rearrangement on the keto-enamine bonds for the strong affinity towards Cu(II).”

2) The authors highlighted the single-atom Cu (II) distribution in TpPa-COF. How to identify the valence state of single-atom Cu except for XPS spectra? It is generally considered that the metal single atom is presented as M (0) anchored on COFs in the references (Chem. Eng. J., 403 (2021) 126383; ACS Catal., 11 (2021) 13266-13279; J. Am. Chem. Soc., 141 (2019) 7615-7621.) What accounts for the difference?

Response: Thanks for the reviewer’s issue.

Except for the XPS and Auger spectra, the valence state of single-atom Cu could be identified by the XANES spectra (*Nat. Commun.* 2022, **13**, 63; *Sci. Adv.* 2022, **8**, eabn9231). The adsorption edge in the region of high binding energy indicates that the complexed metal is in a high valence state. As shown in the XANES spectrum, the adsorption edges of the TpPa-Cu(II) COF and TpPa-Cu(I) COF are both shifted towards the higher binding energy, compared with that of Cu(0) foil. The result demonstrates that the valence of the coordinated Cu metal is positive (*Supplementary Figure 31a*). Furthermore, the edge energy observed from the XANES derivative spectrum of the first peak is typically around 8981 eV for Cu(I) and 8985 eV for Cu(II) ions (*J. Am. Chem. Soc.* 2022, **144**, 10, 4515). Thus, it is known from *Figure R3* that the edge energies at 8984.5 and 8980.9 eV are attributed to the Cu(II) and Cu(I) complexes with TpPa-COF, respectively.

The valence states of single-atom (SA) metals are versatile in the atomically dispersed metal catalysts (*Chem. Rev.* 2020, **120**, 21, 11900). For instance, the SAs in the alloy are presented as M(0), while the SAs stabilized by organic ligands or oxide supports are generally in the cationic state. As reported so far, the COF-based SA catalysts are metal cations such as Cu²⁺, Pt²⁺, and Ni²⁺, all of which are immobilized by the functional groups on the COF frameworks (*Chem. Eng. J.* 2021, **403**, 126383; *ACS Catal.* 2021, **11**, 13266; *J. Am. Chem. Soc.* **2019**, 141, 7615-7621).

The analysis of the valence state for the coordinated Cu(II) is added in the context, **Page 8, Line I-5**, “To our knowledge, when the X-ray absorption near edge structure spectra (XANES) involve a high binding energy of absorption edge, the coordinated metal center is kept in a high-valence state. Thus, compared to the XANES spectrum of Cu foil, it was confirmed that the coordinated Cu metal remained positively charged in the TpPa-COF (**Supplementary Figure 33a**).”

Supplementary Fig. 33 (a) EXAFS and XANES (inset) spectra of the TpPa-Cu(II)-COF, TpPa-Cu(I)-COF, TpPa(Δ)-Cu(I)-COF and Cu foil.

Figure R3. The derivative spectra of the TpPa-Cu(II)-COF, TpPa-Cu(I)-COF, TpPa(Δ)-Cu(I)-COF (calculated from the XANES)

3) From Fig. 1c, the authors mentioned that the other emerging peak at 398.5 eV was ascribed to the newly formed C=N bond. Thus, the authors suggested the occurrence of the keto-to-enol transformation. It is an important conclusion for the proposed structure of the TpPa-Cu(II)-COF complex. However, why the C=N bond is absent in the FTIR spectra?

Response: Thanks for the reviewer's issue.

As the complexation degree in the TpPa-Cu(II) COF is roughly 28%, the C=N vibration band is not apparent in the FT IR spectra. So, the Tp-Cu(II) model complex was prepared for the comparison, and the simulation for the FT IR spectra of the models was performed at the level of PBE0-D3BJ/def2-SVP (**Figure R4**). In the simulated spectra, the C=N band of the Tp-Cu(II) complex appears at 1568 cm^{-1} , which is overlapped with the C=C bands of benzene rings around $1570\text{-}1600\text{ cm}^{-1}$. In the experimental spectra, the C=O band at 1618 cm^{-1} nearly disappears in the spectrum of the Tp-Cu(II) complex. The C=N vibration peak is found at 1601 cm^{-1} , which shifts slightly due to the Cu(II) complexation (*J. Am. Chem. Soc.* 2017, **139**, 17, 6042).

Supplementary Fig. 5 (a) FT IR spectra of the TpPa model and TpPa-Cu(II) models.

Figure R4. The calculated FT-IR spectra of Tp model compound and Tp-Cu(II) model complex.

The change has been made in the main text, *Page 6, Line 15-18*, “The vibration band of C=N was observed at 1601 cm^{-1} , which was coordinated with Cu(II) to induce a slight shift to the low wavenumbers⁴⁹. Also, the C=O peak at 1620 cm^{-1} originating from the TpPa model nearly disappeared in the FT IR spectrum of the TpPa-Cu(II) complex (**Supplementary Fig. 5**).”

4) Some related typical references on the TpPa-COF for photocatalytic hydrogen evolution should be cited in the Introduction section and compared in Table S8, such as *Angew. Chem.* 2018, 130, 1-6; *ACS Catal.*, 11 (2021) 13266-13279.

Response: According to the reviewer’s suggestion, the related references on the TpPa-COF for photocatalytic hydrogen evolution have been cited in the introduction, i.e., *Ref. 39-40*. Also, those reported results have been compared with others in *Supplementary Table 8*. The recommended literature “*Angew. Chem.* 2018, 130, 1-6” cannot be retrieved, and the other “*ACS Catal.*, 11 (2021) 13266-13279” have been cited in the introduction.

The added references are as follows.

39. J. Ming, A. Liu, J. Zhao, et al. Hot π -electron tunneling of metal–insulator–COF nanostructures for efficient hydrogen production. *Angew. Chem. Int. Ed.* 58, 18290 (2019).

40. P. Dong, Y. Wang, et al. Platinum single atoms anchored on a covalent organic framework: boosting active sites for photocatalytic hydrogen evolution. *ACS Catal.* 11, 13266–13279 (2021).

The above two examples are added in *Supplementary Table 8*.

Photocatalysts/Cocatalyst	Irradiation	Sacrificial agents	Solvents	H ₂ evolution rate	Ref.
TpPa-COF/Pt-PVP	>420nm	Ascorbic acid	H ₂ O	8.4 mmol g ⁻¹ h ⁻¹	[35]

TpPa-COF/Pt single-atom	>420nm (265mW cm ⁻²)	Sodium ascorbate	H ₂ O(PBS)	719 μmol g ⁻¹ h ⁻¹	[36]
-------------------------------------	---------------------	-----------------------	--	------

5) In Fig. 2a, why the potential of H⁺/H₂ is presented at pH=5.5? What is the exact potential for cysteine oxidation at the same pH value?

Response: Thanks for the reviewer’s issues.

The potential of H⁺/H₂ is presented at pH = 5.2 as it is the isoelectric point of cysteine. Under this pH condition, the optimum performance of photocatalytic H₂ evolution is achieved with the TpPa(Δ)-Cu(II)-COF(10.39wt%)/L-cysteine (*Supplementary Fig. 36*). When the pH values deviated from 5.2, the HER was dramatically declined. We reason that the neutral cysteine in the isoelectric state is more favorably assembled with the chiral COFs into the enantiomeric mixtures, which allows for the enhanced reaction kinetics in the SED oxidation. **The pH effect on the photocatalytic performance has been discussed in detail in the Q1 of Reviewer #1 (Page R8).**

According to the reviewer’s request, the exact potential for cysteine oxidation at pH = 5.2 was estimated to be -5.41 eV. Thus, *Fig. 2a* and *Supplementary Fig. 24* are revised as shown below. Also, the two potentials are highlighted in the main text, *Page 9, Line 7-10*, “Thus, from the thermodynamic point of view, all the TpPa-Cu(II)-COFs possess the sufficient redox ability for proton reduction (-4.19 eV, pH = 5.2) and cysteine oxidation (-5.41 eV, pH = 5.2) under visible irradiation.”

Figure 2a Energy band structures of TpPa-COF and TpPa-Cu(II)-COFs.

Supplementary Fig. 24 Energy band structures of the achiral TpPa-COF and TpPa-Cu(II)-COF(10.76wt%) and the chiral TpPa-COFs and TpPa-Cu(II)-COFs.

6) In Fig. S15, the signal for the mixture of Cu(II) ions and L-cysteine after 5 min is so weak that it can not be evidence to support the DMPO-trapped sulfanyl radicals. Maybe the authors could extend the irradiation time to detect the signal.

Response: We appreciate the valuable comment from the reviewer.

Herein, *Supplementary Fig. 15* shows the EPR signals of a dark reaction between Cu(II) and cysteine. The result validates that the photocatalysis process involves a dark redox reaction, producing Cu(I) ions and cystine. The reaction intermediate is the sulfonyl radical that can be trapped by DMPO and detected by EPR while the signal intensity is weak. According to the reviewer's suggestion, we supplemented the EPR measurement under light irradiation. With an increase in irradiation time ($\lambda > 420$ nm), the EPR signals are gradually enhanced (*Supplementary Fig. 15b*), reflecting that DMPO-trapped sulfanyl radicals were accumulated.

The related discussion of the EPR test has been revised in the context, *Page 11, Line 12-15*, “During the photocatalysis, the identical radical signals were intensified with the prolonged irradiation as the photoinduced hole on the excited TpPa-Cu(II)-COF was accumulated to enhance the circular conversion between Cu(I) and Cu(II) for the cysteine oxidation (*Supplementary Fig. 15b*).” The supplemented figure is added in the *Supplementary Information, Supplementary Fig. 15b*.

Supplementary Fig. 15 (a) Electron paramagnetic resonance (EPR) spectra of a mixture of L-cysteine (0.15 mol/L), Cu^{2+} aqueous solution (7.5 mmol/L) and DMPO (0.5vol%) recorded at beginning and after 5min at 30°C ($\text{CysS}^{\cdot-}$, marked with *), respectively. (b) EPR spectra of a mixture of TpPa-Cu(II)-COF (100 mg/L), L-cysteine (0.1 mol/L) and DMPO (0.5vol%) collected with irradiation, $\lambda > 420 \text{ nm}$ ($\text{CysS}^{\cdot-}$, marked with *).

7) In-Page 11, the authors mentioned that electrons transfer from the Cu(I) donor to the photogenerated acceptor on the excited TpPa-COF upon visible irradiation. Why is it not from the excited TpPa-COF to the Cu(I)? How to confirm the electron's transfer direction?

Response: Thanks for the reviewer's issue.

The electron's transfer direction is determined by time-dependent DFT calculation. Taking all the relevant orbitals into account, we analyzed the difference in the wavefunctions before and after excitation on the Tp-Cu(I) model. **Fig. 2e** displays the real-space electron-hole distribution of the model, wherein the blue regions are dominated by holes on the Tp-Cu(I) complex, and the green regions on the Tp moiety accumulate electrons. It means that the photo-induced electron transfer from the excited Tp-Cu(I) complex to the excited Tp moiety causes the generation of the electron-deficient region (blue) and electron-rich region (green). Meanwhile, based on the inter-fragment charge transfer (IFCT) method, the photo-induced electron depletion on Cu(I) was determined to be 0.32 e (net). It can be seen that the net electron outflowing (the iso-surface with a single color) from the Cu(I) atom in **Fig. 2e**.

The detailed description for the real-space electron-hole distribution has been supplemented in the main text, **Page 13, Line 1-6**, "**Fig. 2e** displays the difference in the wavefunctions before and after vertical excitation on the model, in which the green region mainly at the Tp moiety signifies the photo-induced electron accumulation and the blue region at the Tp-Cu(I) complex illustrates the photo-induced electron depletion. Therefore, it is ascertained that the excited-state electrons transfer from the Tp-Cu(I) complex to the Tp moiety via the phenyl linker, resulting in the rearrangement of electron-hole distribution in the excited state."

8) In Fig. S26, the authors completed the photocatalytic recycling performances of TpPa(Δ)-Cu(II)-COF under 24-h irradiation of visible light. They mentioned that the recycled sample was washed with 1M HCl to remove the remaining Cu(II) ions after one cycle of 6-h photocatalysis, and loaded with similar content of fresh Cu(II) ions for the next photocatalytic cycle. Does it mean the Cu(II) ions are easily removed during the photocatalytic process for the sample of TpPa(Δ)-Cu(II)-COF? If that, the Cu(II) ions in TpPa(Δ)-Cu(II)-COF is not stable. The authors should explain it in detail.

Response: Thanks for the reviewer's issue.

During the photocatalytic reaction, the photo-oxidation of cysteine produced cystine to precipitate into the solution at pH = 5.2. After the reaction, the solid collected from the solution was a mixture of TpPa-Cu(II)-COF and cystine (*Supplementary Fig. 28*). To purify the recycled COF solid, the mixture was rinsed with 1M HCl aqueous solution to dissolve cystine, but the complexes between Cu(II) and TpPa-COF completely decomposed (99%, measured by ICP). Therefore, the recycled TpPa-COF solid needed to be coordinated with Cu(II) again for the next photocatalytic cycle. Under the photocatalytic conditions, a majority of the coordinated Cu (70~80%) remained as the TpPa(Δ)-Cu(II)-COF complex was relatively stable at pH = 5.2.

The explanation has been made in the context, *Page 16, Line 8-13*, “After every cycle, the collected photocatalysts were rinsed with 1M HCl to eliminate cystine as it precipitated out and mixed with the COF solid (*Supplementary Fig. 28*). Although the acid treatment caused the coordination decomposition between TpPa(Δ)-COF and Cu(II), the obtained TpPa-COF solid could be re-coordinated with the equal quantity of Cu(II) for the following photocatalytic cycle.”

Supplementary Fig. 28 Photographs of the reaction solution before and after 6-h photocatalytic reaction.

Reviewer #3 (Remarks to the Author):

The manuscript report the enantioselective combination of a chiral ketoenamine-linked COF with L-/D-cysteine as the sacrificial electron donor for hydrogen evolution. Without precious 22 metal co-catalysts, the photocatalytic enantiomers can significantly enhance the H₂ evolution rate of up to 14.72 mmol h⁻¹g⁻¹ with a high sacrificial oxidation turnover frequency of 9.0 h⁻¹. The origin of superior performance lies in the increase in the reaction kinetics. There some problems need be addressed.

1. The chiral induced spin selectivity (CISS) effect has been well studied in nanostructured inorganic and polymeric materials (eg *Acc. Chem. Res.* 2020, 53, 2659–2667). Introducing chirality into metal-semiconductor hybrid nanostructures can boost the chiral hot electrons or spin-selective electrons of plasmonic nanocomponents, which can transfer to a catalytic semiconductor and trigger asymmetric catalytic reactions. Very recently, Wang et al reported that the chiral hybrid nanostructures can drive chirality-dependent photocatalytic hydrogen generation (doi.org/10.1002/anie.202112400). Unfortunately, all these previous reports are not mentioned in the manuscript.

Response: According to the reviewer's suggestion, we have cited the relevant literatures reporting on the CISS effect for spin-controlled catalysis (*Acc. Chem. Res.* 2020, 53, 2659) and the recent study on improving the HER performance of chiral hybrid nanostructures under polarized irradiation (*Angew. Chem. Int. Ed.* 2021, 61, e202112400).

The CISS effect has been described in the context, **Page 3, Line 11-14**, “More intriguingly, chiral-induced spin selectivity and polarization contribute to elevating the intermediate reactivity and selecting the desired reaction pathways, e.g., oxygen evolution reaction in water electrolysis⁵ and photocatalytic water splitting under circularly polarized light⁶.”

The added references are as follows.

5. R. Naaman, Y. Paltiel, D. H. Waldeck, Chiral Induced Spin Selectivity Gives a New Twist on Spin-Control in Chemistry. *Acc. Chem. Res.* **53**, 2659-2667 (2020).

6. L. Tan, S.-J. Yu, Y. Jin, et al. Inorganic Chiral Hybrid Nanostructures for Tailored Chiroptics and Chirality-Dependent Photocatalysis. *Angew. Chem. Int. Ed.* **61**, e202112400 (2021).”

2. Strangely, the structure and PHE of the achiral COF with copper salt were well studied and discussed, but those of the chiral COF were even not provided. It is difficult to follow the main topic of this manuscript.

Response: Thanks for the reviewer's suggestion. Here, we would like to highlight all our studies on the chiral COFs to be easily accessed for Reviewer #3.

- (1) We have systematically investigated the composition, structure, porosity, and chirality of the chiral TpPa(Δ)-COF by FT IR, PXRD, N₂ isotherm sorption and CD measurements, respectively, in **Fig. 3a, 3c, and Supplementary Fig. 1b, 1e-1h, 3b**.
 - (2) The coordination of the chiral TpPa-COF with Cu species was characterized by XPS, ICP, and XANES, respectively, in **Supplementary Fig. 3c-3d, 14b-14c, 23, 33-34 and Supplementary Table 1, 7**.
 - (3) The enantiomeric interaction between the chiral TpPa-COF and cysteine was estimated by the ITC test and theoretical calculation in **Fig. 4 and Supplementary Fig.35**.
 - (4) The photocatalytic performance of the chiral TpPa-Cu(II)-COF was evaluated in conjunction with the different chiral SEDs in **Fig. 3d**.
 - (5) The long-term catalytic stability and post-photocatalysis structure of the chiral TpPa-Cu(II)-COF was tested in **Supplementary Fig. 27, 29**.
 - (6) The AQEs and TOF_{ox} were assessed for the enantiomeric mixture between chiral COFs and SEDs in **Fig. 3e and Supplementary Fig. 26**.
 - (7) The photophysical studies, including UV-vis, photocurrent, Nyquist plots, Mott-Schottky tests, and photoluminescent lifetimes, were conducted for the chiral TpPa-Cu(II)-COF in **Supplementary Fig. 8-10, 30-32**.
 - (8) The energy band structures of the different chiral TpPa-COF were established in **Supplementary Fig. 24**.
 - (9) The Tafel-polarization curves of the chiral TpPa-COF were displayed in **Fig. 3f**.
 - (10) The catalytic mechanism of the chiral TpPa-COF was proposed as shown in **Fig. 5c,5d**.
- All the above details of Items 1-10 can be found in the main text, Page 14-19, Section: Enantioselective combination of chiral TpPa-Cu(II)-COF with cysteine for photocatalytic H₂ evolution.*

In response to all the suggestions and issues of Reviewer #1 and Reviewer #2, we have supplemented the experiments and characterizations for the chiral COFs.

- (1) The photocatalytic performances were tested for the chiral TpPa-Cu(II)-COF synthesized in the different batches, **Supplementary Fig. 25**.
- (2) The pH effect on the HERs of the chiral TpPa(Δ)-Cu(II)-COF was studied, **Supplementary Fig. 36**.
- (3) L-/D-ascorbic acid was used as SEDs to assembly with the chiral TpPa-COF for the photocatalytic H₂ evolution with or without Pt as co-catalysts, **Supplementary Fig. 38-41**.
- (4) The other revised data, including energy band structures and CV for the chiral TpPa-Cu(II)-COF, were shown in **Supplementary Fig. 16, 24**.

All the above details of Items 1-4 can be found in response to Reviewer #1 and Reviewer #2, Page R1-R21. Also, the manuscript has been revised with the supplemented experiments.

So, although this work is interesting, I cannot recommend its publication.

REVIEWERS' COMMENTS

Reviewer #1 (Remarks to the Author):

In this revised version of the manuscript entitled “Enantioselective Combination of Chiral Covalent Organic Frameworks ~”, the authors have addressed all my concerns well in particular for the requests of some additional experimental results. Still I am a bit doubtful on the role of Cu in photocatalytic activity denying the activity of the coordinated Cu ions, but the revised discussions as well as the supporting data are convincing enough.

The results on the activity with L- and D-ascorbic acids are as I expected, and suggesting the spin selective reactions mediated by electrons transporting chiral COF platforms. We are also working on the similar systems, but you got ahead of us! Congratulation.

I feel now the manuscript is ready for publication in Nature Communications.

Reviewer #2 (Remarks to the Author):

This manuscript has been carefully revised according to reviewers' comments. It should be accepted.

Reviewer #3 (Remarks to the Author):

This revised manuscript can be published in Nature Communication.

REVIEWERS' COMMENTS

Reviewer #1 (Remarks to the Author):

In this revised version of the manuscript entitled “Enantioselective Combination of Chiral Covalent Organic Frameworks ~”, the authors have addressed all my concerns well in particular for the requests of some additional experimental results. Still I am a bit doubtful on the role of Cu in photocatalytic activity denying the activity of the coordinated Cu ions, but the revised discussions as well as the supporting data are convincing enough.

The results on the activity with L- and D-ascorbic acids are as I expected, and suggesting the spin selective reactions mediated by electrons transporting chiral COF platforms. We are also working on the similar systems, but you got ahead of us! Congratulation.

I feel now the manuscript is ready for publication in Nature Communications.

Response: Thanks for the reviewer’s recommendation and suggestion on exploring chiral COFs for spin-selective reactions in the following study. As for the role of Cu in photocatalysis, it has been evidenced that the coordinated Cu(II) ions are not the predominated active catalytic sites as they can only be converted into Cu(I) ions instead of Cu(0) atoms in the presence of cysteine. The formed Cu(I) ions are insufficient to drive a hydrogen reduction reaction. Therefore, the role of Cu in the current work is evident, mainly working as an electron transfer mediator during photocatalysis. If SED has a stronger reductibility, the coordinated Cu(II) ions may be reduced to Cu(0) atoms for direct hydrogen production.

Reviewer #2 (Remarks to the Author):

This manuscript has been carefully revised according to reviewers' comments. It should be accepted.

Response: Thanks for the reviewer’s recommendation.

Reviewer #3 (Remarks to the Author):

This revised manuscript can be published in Nature Communication.

Response: Thanks for the reviewer’s recommendation.